# Causal Effect Identification in a Sub-Population with Latent Variables

**Amir Mohammad Abouei** [1], **Ehsan Mokhtarian** [1], **Negar Kiyavash** [2], **Matthias Grossglauser** [1]

[1]School of Computer and Communication Sciences, EPFL
[2]College of Management of Technology, EPFL
{amir.abouei, ehsan.mokhtarian, negar.kiyavash, matthias.grossglauser}@epfl.ch

## Abstract

The S-ID problem seeks to compute a causal effect in a specific sub-population from the observational data pertaining to the same sub-population [AMK24]. This problem has been addressed when all the variables in the system are observable. In this paper, we consider an extension of the S-ID problem that allows for the presence of latent variables. To tackle the challenges induced by the presence of latent variables in a sub-population, we first extend the classical relevant graphical definitions, such as C-components and Hedges, initially defined for the so-called ID problem [Pea95, TP02], to their new counterparts. Subsequently, we propose a sound algorithm for the S-ID problem with latent variables.

## 1 Introduction

Causal inference, i.e., understanding the effect of an intervention in a stochastic system, is a key focus of research in statistics and machine learning [Rub74, Pea00, Pea09, SGSH00]. Scientists, policymakers, business leaders, and healthcare professionals must understand causal relationships to move beyond correlations and make informed, evidence-based decisions. To perform causal inference tasks, it is crucial to differentiate between two types of data: observational and interventional [PM18].

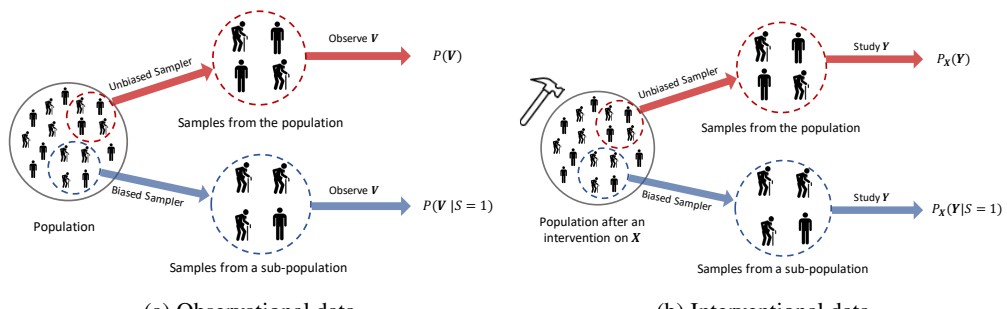

(a) Observational data.        (b) Interventional data.

Figure 1: A population consists of a sample space for the study of the causal effect of an intervention. While the unbiased sampler draws samples uniformly at random, the biased sampler selects samples based on certain criteria, forming a sub-population.

**Observational Data.** Figure 1 illustrates a *population* that pertains to the entire *sample space* for a study of the causal effect of some intervention. A *sampler* draws samples from the population. The sampler is unbiased if it draws samples at random such that each individual in the population has

Table 1: Various causal effect identification problems. $\mathbf{X}$ is the set of intervened variables, $\mathbf{Y}$ is the set of outcome variables, and $S = 1$ corresponds to a sub-population. ID, c-ID, and s-Recoverability have been addressed in the presence of latent variables. s-ID problem has only been studied in causally sufficient cases where all variables are observed.

| Problem | Given distribution | Target distribution | Presence of latent variables |
|---|---|---|---|
| ID | $P(\mathbf{V})$ | $P_\mathbf{X}(\mathbf{Y})$ | ✓ |
| c-ID | $P(\mathbf{V})$ | $P_\mathbf{X}(\mathbf{Y}|\mathbf{Z})$ | ✓ |
| s-Recoverability | $P(\mathbf{V}|S = 1)$ | $P_\mathbf{X}(\mathbf{Y})$ | ✓ |
| s-ID | $P(\mathbf{V}|S = 1)$ | $P_\mathbf{X}(\mathbf{Y}|S = 1)$ | ✗ |

an equal chance of being selected. As a result, the obtained sample is representative of the entire population. In contrast, a biased sampler selects samples based on certain criteria forming a *sub-population*. For each extracted sample, we collect data from a set of observed features denoted by $\mathbf{V}$. As depicted in Figure 1a, when the sampler is unbiased, the observational data comes from the joint distribution $P(\mathbf{V})$. For a biased sampler, the observations can be modeled as drawn from a conditional distribution $P(\mathbf{V}|S = 1)$, where $S = 1$ indicates that the sample belongs to a sub-population.

**Interventional Data.** An *intervention* on a subset $\mathbf{X} \subseteq \mathbf{V}$ assigns specific values to the variables in the subset. If performing an intervention results in changes in other variables of interest, it suggests a causal relationship, apart from mere correlation. Interventions are often represented with the $do()$ operator, highlighting the deliberate change of a variable [Pea00, Pea09]. For the sake of simplicity in notation, we use $P_\mathbf{X}(\cdot)$ to denote the distribution of the variables after an intervention on $\mathbf{X}$. Figure 1b depicts the population after an intervention on subset $\mathbf{X} \subseteq \mathbf{V}$, where we seek to understand how changes in $\mathbf{X}$ would affect a set of outcome variables $\mathbf{Y} \subseteq \mathbf{V} \setminus \mathbf{X}$. To analyze this causal effect across the entire population, we must compute the distribution $P_\mathbf{X}(\mathbf{Y})$. On the other hand, if we are merely interested in the results of the intervention on a specific sub-population, it suffices to compute the conditional distribution $P_\mathbf{X}(\mathbf{Y}|S = 1)$ pertaining to the sub-population.

**Causal Effect Identification.** Performing interventions in populations can be challenging due to high costs, ethical concerns, or sheer impracticability. Instead, researchers often use observational methods, leveraging the environment's *causal graph*, a graphical representation that depicts the causal relationships between variables [Pea09, SGSH00], and observational data to estimate interventional distributions of interest. Various *causal effect identification* problems in the causal inference literature are concerned with this issue.

**Related Work.** Table 1 lists four causal effect identification problems. The most renowned among them is the ID problem, introduced by [Pea95], which seeks to determine a causal effect for the entire population using the observational distribution pertaining to the entire population. Specifically, it aims to compute $P_\mathbf{X}(\mathbf{Y})$ from $P(\mathbf{V})$. The c-ID problem, introduced by [SP06a], extends the ID problem to handle conditional causal effects, i.e., compute the conditional causal effect $P_\mathbf{X}(\mathbf{Y}|\mathbf{Z})$ from the observational distribution $P(\mathbf{V})$ pertaining to the entire population. [BTP14] introduced the s-Recoverability problem that focuses on inferring the causal effect of $\mathbf{X}$ on $\mathbf{Y}$ for the entire population using data drawn solely from a specific sub-population. [AMK24] introduced s-ID, which asks whether a causal effect in a sub-population such as $P_\mathbf{X}(\mathbf{Y}|S = 1)$ can be uniquely computed from the observational distribution pertaining to that sub-population, i.e., $P(\mathbf{V}|S = 1)$. Another direction of research considers learning a causal effect from multiple datasets [LCB19, KMEK22, CLB21, KEK23, THK21, LGS24]. In all aforementioned causal inference problems, the causal graph is assumed to be known. Some recent work relax this assumption [JZB19, JRZB22] or introduce additional conditions on the causal graph with the goal of identifying a broader range of causal effects [THK19, MJEK22, JAK23]. Settings where data samples are dependent introduce new challenges to causal inference, which have been explored in another line of research [SS18, BMS20, ZMP23].

**s-ID Is Not ID.** It is worth emphasizing that the s-ID problem is not a special case of ID problem, where the population is restricted to the target sub-population. The presence of selection bias $S$ introduces additional dependencies among variables, and ignoring $S$ in the graph invalidates the application of the rules of do-calculus [Pea00] (which are the main tools used to tackle the ID problem) on input distribution, i.e., $P(\mathbf{V}|S = 1)$. Consequently, there are many instances where a causal effect is identifiable in the ID setting but not identifiable in the s-ID setting. In particular, when all the variables in a causal system are observable, all causal effects are identifiable in the setting of

the ID problem [Pea00]. This is not the case in the S-ID setting, and some causal effects become non-identifiable, as noted by [AMK24]. Moreover, even when a causal effect is identifiable in both the ID and S-ID settings, using the expression from the ID algorithm can lead to erroneous inference. Example 1 illustrates this case.

**Example 1.** Consider an example pertaining to study of the effect of a cholesterol-lowering medication on cardiovascular disease. Figure 2 depicts the causal graph of this example, where $X$ is the medication choice that directly affects $Y$, cardiovascular disease. Variable $Z$ represents the diet and exercise routine of a person. In this scenario, $X$ and $Z$ are confounded by the person's socioeconomic status (e.g., income). It can be shown that the causal effect of $X$ on $Y$ (in the entire population, for instance, the people around the globe) is identifiable (ID) from $P(X, Y, Z)$ and can be computed as $P_X(Y) = P(Y|X)$.

However, we might instead be interested in a study that focuses on the people of a specific region. In this case, the target sub-population would correspond to individuals who are biased toward particular diet and exercise routines and possibly have a higher genetic predisposition for heart disease. Let $S$ be an indicator node for this sub-population, $Z$ has a directed edge toward $S$, and $S$ and $Y$ are confounded by the latent genetic predisposition of the people of this group. We will show in Section 5 that the causal effect in this sub-population, $P_X(Y|S = 1)$, is S-ID and equals $\sum_Z P(Y|X, Z, S = 1)P(Z|S = 1)$. In

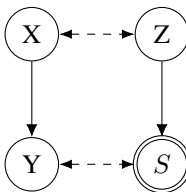

Figure 2: ADMG $\mathcal{G}^{\mathrm{s}}$ in Example 1.

this example, the presence of $S$ introduces a spurious correlation between $X$ and $Y$ through the path involving $Z$ and $S$. Therefore, if we were to ignore the presence of $S$ and apply the ID algorithm to the input $P(X, Y, Z|S = 1)$, it would result in incorrect inference: $P(Y|X, S = 1)$ as opposed to the correct value $P_X(Y|S = 1)$. In Appendix D, we empirically compare the differences between ID and S-ID settings. We present another example in Appendix A where a causal effect is ID but not S-ID.

In this paper, we consider the S-ID problem in the presence of latent variables. Our main contributions are as follows.

- We extend the classical relevant graphical definitions, such as c-components and Hedges, initially defined for the ID problem so that they inherit the key properties of their predecessors but are applicable to the S-ID setting in the presence of latent variables (Section 4).

- We present a sufficient graphical condition to determine whether a causal effect is S-ID (Theorem 5.1). Accordingly, we propose a sound algorithm for the S-ID problem (Algorithm 1).

- We show a reduction from the S-Recoverability problem to the S-ID problem (Theorem 6.1), indicating that solving S-ID can also solve the S-Recoverability problem.

**Organization.** In Sections 2 and 3, we cover the preliminaries and review key definitions and results for the ID problem. In Section 4, we formally define the S-ID problem in the presence of latent variables and present the proper modifications of the classical graphical notions of interest for the S-ID problem. We present our main results in Section 5. In Section 6, we introduce a reduction from S-Recoverability to S-ID. The appendix includes proofs of our results, as well as a numerical experiment.

## 2   Preliminaries

Throughout the paper, we use capital letters to represent random variables and bold letters to represent sets of variables. Furthermore, to facilitate ease of reading, we have summarized the key notations in Table 2.

**Graph Definitions.** Acyclic directed mixed graphs (ADMGs) consist of a mix of directed and bidirected edges and have no directed cycles. Let $\mathcal{G} = (\mathbf{V}, \mathbf{E}_1, \mathbf{E}_2)$ be an ADMG, where $\mathbf{V}$ is a set of variables, $\mathbf{E}_1$ is a set of directed edges ($\rightarrow$), and $\mathbf{E}_2$ is a set of bidirected edges ($\leftrightarrow$). The set of parents of a variable $X \in \mathbf{V}$, denoted by $Pa_{\mathcal{G}}(X)$, consists of the variables with a directed edge to $X$. Similarly, the set of ancestors of $X \in \mathbf{V}$, denoted by $Anc_{\mathcal{G}}(X)$, includes all variables on a

Table 2: Table of notations.

| Notation | Description |
|---|---|
| $\mathbf{V}, \mathbf{U}$ | Sets of observed and unobserved variables |
| $S$ | Auxiliary vertex (variable) used to model a sub-population |
| $\mathcal{G}^{\mathrm{s}}$ | Augmented ADMG over $\mathbf{V} \cup \{S\}$ |
| $Pa_{\mathcal{G}}(X)$ | Parents of vertex $X$ in graph $\mathcal{G}$ |
| $Anc_{\mathcal{G}}(X), Anc_{\mathcal{G}}(\mathbf{X})$ | Ancestors of vertex $X$ (including $X$); the union of ancestors for all $X \in \mathbf{X}$ |
| $\mathbf{V}_{\mathrm{AS}}, \mathbf{V}_{\mathrm{NS}}$ | $\mathbf{V} \cap Anc_{\mathcal{G}^{\mathrm{s}}}(S)$ and its complement, $\mathbf{V} \setminus Anc_{\mathcal{G}^{\mathrm{s}}}(S)$ |
| $\mathcal{G}[\mathbf{X}]$ | Subgraph of $\mathcal{G}$ induced by the vertices in $\mathbf{X}$ |
| $\mathcal{G}_{\overline{\mathbf{X}}\underline{\mathbf{Z}}}$ | Subgraph of $\mathcal{G}$ after removing incoming edges to $\mathbf{X}$ and outgoing edges from $\mathbf{Z}$ |
| $P^{\mathrm{s}}(\mathbf{V})$ | Sub-population distribution, i.e., $P(\mathbf{V}|S=1)$ |
| $P_{\mathbf{X}}(\mathbf{Y})$ | Causal effect of $\mathbf{X}$ on $\mathbf{Y}$, i.e., post-interventional distribution |
| $P^{\mathrm{s}}_{\mathbf{X}}(\mathbf{Y})$ | Causal effect of $\mathbf{X}$ on $\mathbf{Y}$ in the sub-population, i.e., $P_{\mathbf{X}}(\mathbf{Y}|S=1)$ |
| $Q[\mathbf{H}]$ | $P_{\mathbf{V} \setminus \mathbf{H}}(\mathbf{H})$ |
| $Q^{\mathrm{s}}[\mathbf{H}]$ | $P_{\mathbf{V}_{\mathrm{NS}} \setminus \mathbf{H}}(\mathbf{H}|Anc_{\mathcal{G}^{\mathrm{s}}}(S) \setminus \{S\}, S=1), \forall \mathbf{H} \subseteq \mathbf{V}_{\mathrm{NS}}$ |

*directed* path to $X$, including $X$ itself. For a set $\mathbf{X} \subseteq \mathbf{V}$, we define $Anc_{\mathcal{G}}(\mathbf{X}) = \bigcup_{X \in \mathbf{X}} Anc_{\mathcal{G}}(X)$. In an ADMG $\mathcal{G}$ over $\mathbf{V}$, a subset $\mathbf{X} \subseteq \mathbf{V}$ is called ancestral if $Anc_{\mathcal{G}}(\mathbf{X}) = \mathbf{X}$.

A path is called bidirected if it only consists of bidirected edges. A non-endpoint vertex $X_i$ on a path $(X_1, X_2, \ldots, X_k)$ is called a *collider* if one of the following situations arises:

$$X_{i-1} \rightarrow X_i \leftarrow X_{i+1}, \quad X_{i-1} \leftrightarrow X_i \leftarrow X_{i+1}, \quad X_{i-1} \rightarrow X_i \leftrightarrow X_{i+1}, \quad X_{i-1} \leftrightarrow X_i \leftrightarrow X_{i+1}.$$

Let $\mathbf{X}, \mathbf{Y}, \mathbf{W}$ be three disjoint subsets of variables in an ADMG $\mathcal{G}$. A path $\mathcal{P} = (X, Z_1, \ldots, Z_k, Y)$ between $X \in \mathbf{X}$ and $Y \in \mathbf{Y}$ in $\mathcal{G}$ is called *blocked* by $\mathbf{W}$ if there exists $1 \leq i \leq k$ such that $Z_i$ is a collider on $\mathcal{P}$ and $Z_i \notin Anc_{\mathcal{G}}(\mathbf{W})$, or $Z_i$ is not a collider on $\mathcal{P}$ and $Z_i \in \mathbf{W}$.

Denoted by $(\mathbf{X} \perp\!\!\!\perp \mathbf{Y}|\mathbf{W})_{\mathcal{G}}$, we say $\mathbf{W}$ $m$-separates $\mathbf{X}$ and $\mathbf{Y}$ if for any $X \in \mathbf{X}$ and $Y \in \mathbf{Y}$, $\mathbf{W}$ blocks all the paths in $\mathcal{G}$ between $X$ and $Y$. Conversely, $(\mathbf{X} \not\perp\!\!\!\perp \mathbf{Y}|\mathbf{W})_{\mathcal{G}}$ if there exists at least one path between a variable in $\mathbf{X}$ and a variable in $\mathbf{Y}$ that is not blocked by $\mathbf{W}$.

For $\mathbf{X}, \mathbf{Z} \subseteq \mathbf{V}$, $\mathcal{G}_{\overline{\mathbf{X}}\underline{\mathbf{Z}}}$ denotes the edge subgraph of $\mathcal{G}$ obtained by removing the edges with an arrowhead to a variable in $\mathbf{X}$ (including bidirected edges) and outgoing edges of $\mathbf{Z}$ (excluding bidirected edges). Moreover, $\mathcal{G}[\mathbf{X}]$ denotes the vertex subgraph of $\mathcal{G}$ consisting of $\mathbf{X}$ and bidirected and directed edges between them.

**SCM.** A structural causal model (SCM) is a tuple $(\mathbf{V}, \mathbf{U}, \mathbf{F}, P(\mathbf{U}))$, where $\mathbf{V}$ is a set of endogenous variables, $\mathbf{U}$ is a set of exogenous variables independent from each other with the joint probability distribution $P(\mathbf{U})$, and $\mathbf{F} = \{f_X\}_{X \in \mathbf{V}}$ is a set of deterministic functions such that for each $X \in \mathbf{V}$,

$$X = f_X(Pa^X, \mathbf{U}^X),$$

where $Pa^X \subseteq \mathbf{V} \setminus \{X\}$ and $\mathbf{U}^X \subseteq \mathbf{U}$. This SCM induces a causal graph $\mathcal{G}$ over $\mathbf{V}$ such that $Pa_{\mathcal{G}}(X) = Pa^X$ and there is a bidirected edge between two distinct variables $X, Y \in \mathbf{V}$ when $\mathbf{U}^X \cap \mathbf{U}^Y \neq \varnothing$. Henceforth, we assume the underlying SCM induces a causal graph that is ADMG, i.e., it contains no directed cycles.

An SCM $\mathcal{M} = (\mathbf{V}, \mathbf{U}, \mathbf{F}, P(\mathbf{U}))$ with causal graph $\mathcal{G}$ induces a unique joint distribution over the variables $\mathbf{V}$ that can be factorized as

$$P^{\mathcal{M}}(\mathbf{V}) = \sum_{\mathbf{U}} \prod_{X \in \mathbf{V}} P^{\mathcal{M}}(X|Pa_{\mathcal{G}}(X)) \prod_{U \in \mathbf{U}} P^{\mathcal{M}}(U).$$

This property is known as the Markov factorization [Pea09]. Note that $\sum_{\mathbf{X}}$ denotes marginalization, i.e., summation (or integration for continuous variables) over all the realizations of the variables in set $\mathbf{X}$. We often drop the $\mathcal{M}$ in $P^{\mathcal{M}}(\cdot)$ when it is clear from the context.

**Modeling a Sub-Population.** We model a sub-population using an auxiliary variable $S$ and a biased sampler from a causal environment akin to [BP12, AMK24]. Suppose $\mathcal{M}$ is the underlying SCM of an environment with the set of observed variables $\mathbf{V}$. In this causal environment, an unbiased sampler produces samples drawn from $P(\mathbf{V})$. When the sampler is biased, it draws samples from the conditional distribution $P^{\mathrm{s}}(\mathbf{V}) := P(\mathbf{V}|S=1)$, where $S$ is an auxiliary variable defined as

$S := f_S(\text{Pa}^S, \mathbf{U}^S)$, where $f_S$ is a binary function, $\text{Pa}^S \subseteq \mathbf{V}$, and $\mathbf{U}^S$ is the set of exogenous variables corresponding to $S$. Note that $\mathbf{U}^S$ can intersect with $\mathbf{U}$, but the variables in $\mathbf{U} \cup \mathbf{U}^S$ are assumed to be independent. In this model, $S = 1$ indicates that the sample is drawn from the target sub-population. Furthermore, we define the augmented SCM $\mathcal{M}^s = (\mathbf{V} \cup \{S\}, \mathbf{U} \cup \mathbf{U}^S, \mathbf{F} \cup \{f_S\}, P(\mathbf{U} \cup \mathbf{U}^S))$ obtained by adding $S$ to the underlying SCM $\mathcal{M}$. We denote by $\mathcal{G}^s$, the causal graph of $\mathcal{M}^s$, which is an augmented ADMG over $\mathbf{V} \cup \{S\}$. Note that in $\mathcal{G}^s$, variable $S$ does not have any children, but it can have several parents and bidirected edges.

**Intervention.** An *intervention* on a set $\mathbf{X} \subseteq \mathbf{V}$ converts $\mathcal{M}$ to a new SCM where the equations of the variables in $\mathbf{X}$ are replaced by some constants. We denote by $Q[\mathbf{V} \setminus \mathbf{X}] := P_{\mathbf{X}}(\mathbf{V} \setminus \mathbf{X})$ the corresponding post-interventional distribution, i.e., the joint distribution of the variables in the new SCM. The causal effect of $\mathbf{X}$ on $\mathbf{Y}$ refers to the post-interventional distribution $P_{\mathbf{X}}(\mathbf{Y})$, where $\mathbf{X}$ and $\mathbf{Y}$ are disjoint subsets of $\mathbf{V}$. Accordingly, the causal effect of $\mathbf{X}$ on $\mathbf{Y}$ in a sub-population is denoted by $P_{\mathbf{X}}(\mathbf{Y}|S = 1)$.[1]

**Problem Setup.** Let $(\mathbf{V}, \mathbf{U}, \mathbf{F}, P(\mathbf{U}))$ be an SCM with ADMG $\mathcal{G}$ representing its causal graph. Additionally, let $S$ be an auxiliary variable representing a specific sub-population. In this paper, given the augmented graph $\mathcal{G}^s$ and two arbitrary, disjoint subsets $\mathbf{X}$ and $\mathbf{Y}$, we address the following question: Can the causal effect $P_{\mathbf{X}}^s(\mathbf{Y})$ be uniquely identified from the observational distribution $P^s(\mathbf{V})$? Please refer to Definition 4.1 for the formal definition of the s-ID problem.

# 3  ID, C-component, and Hedge

Our proposed approach to address the s-ID problem extends certain definitions and properties from the classic ID problem [Pea95]. For the sake of completeness and pedagogical reasons, in this section, we review some definitions and the main results in the ID problem [TP02, HV06, SP06b].

**Definition 3.1** (ID). Suppose $\mathcal{G}$ is an ADMG over $\mathbf{V}$ and let $\mathbf{X}$ and $\mathbf{Y}$ be disjoint subsets of $\mathbf{V}$. Causal effect $P_{\mathbf{X}}(\mathbf{Y})$ is said to be identifiable (or ID for short) in $\mathcal{G}$ if for any two SCMs $\mathcal{M}_1$ and $\mathcal{M}_2$ with causal graph $\mathcal{G}$ for which $P^{\mathcal{M}_1}(\mathbf{V}) = P^{\mathcal{M}_2}(\mathbf{V}) > 0$, then $P_{\mathbf{X}}^{\mathcal{M}_1}(\mathbf{Y}) = P_{\mathbf{X}}^{\mathcal{M}_2}(\mathbf{Y})$.

Next, we review C-components, a fundamental concept to address the ID problem.

**Definition 3.2** (C-component). Suppose $\mathcal{G}$ is an ADMG over $\mathbf{V}$. The C-components of $\mathcal{G}$ are the connected components in the graph obtained by considering only the bidirected edges of $\mathcal{G}$. Furthermore, $\mathcal{G}$ is called a single C-component if it contains only one C-component.

There exist a few different definitions for *Hedge*, another central notion in the ID literature. Here, we provide a somewhat simplified definition that not only suffices to present the main result of the ID problem but also allows us to extend it in the next section to the s-ID setting.

**Definition 3.3** (Hedge). Suppose $\mathcal{G}$ is an ADMG over $\mathbf{V}$, and let $\mathbf{Y} \subseteq \mathbf{V}$ such that $\mathcal{G}[\mathbf{Y}]$ is a single C-component. A subset $\mathbf{H} \subseteq \mathbf{V}$ is called a Hedge for $\mathbf{Y}$ in $\mathcal{G}$, if $\mathbf{Y} \subsetneq \mathbf{H}$, $\mathcal{G}[\mathbf{H}]$ is a single C-component, and $\mathbf{H} = Anc_{\mathcal{G}[\mathbf{H}]}(\mathbf{Y})$.

**Example 2.** Consider the ADMG $\mathcal{G}$ over $\mathbf{V} = \{X_1, X_2, Y_1, Y_2\}$ depicted in Figure 3a. In this case, $\mathcal{G}$, $\mathcal{G}[X_1, X_2]$, and $\mathcal{G}[Y_1, Y_2]$ are single C-components. The C-components of $\mathcal{G}[X_1, X_2, Y_1]$ are $\{X_1, X_2\}$ and $\{Y_1\}$. Furthermore, $\mathbf{H} = \{X_1, Y_1, Y_2\}$ is a Hedge for $\mathbf{Y} = \{Y_1, Y_2\}$ since $\mathcal{G}[\mathbf{H}]$ and $\mathcal{G}[\mathbf{Y}]$ are single C-components and $Anc_{\mathcal{G}[\mathbf{H}]}(\mathbf{Y}) = \mathbf{H}$. Similarly, $\mathbf{V}$ is a Hedge for $\mathbf{Y}$, but there exists no Hedge for either $\{Y_1\}$ or $\{Y_2\}$.

The following theorem, restating the results in [SP06b] and [HV06], outlines a necessary and sufficient condition to determine the identifiability of a causal effect in an ADMG.

**Theorem 3.4** (ID). *Let $\mathcal{G}$ be an ADMG over $\mathbf{V}$, and $\mathbf{X}$ and $\mathbf{Y}$ be two disjoint subsets of $\mathbf{V}$. Causal effect $P_{\mathbf{X}}(\mathbf{Y})$ is ID in $\mathcal{G}$ if and only if $Q[\mathbf{D}]$ is ID in $\mathcal{G}$, where $\mathbf{D} = Anc_{\mathcal{G}[\mathbf{V} \setminus \mathbf{X}]}(\mathbf{Y})$. Furthermore, let $\{\mathbf{D}_i\}_{i=1}^k$ be the C-components of $\mathcal{G}[\mathbf{D}]$, then $Q[\mathbf{D}]$ is ID in $\mathcal{G}$ if and only if there are no Hedge in $\mathcal{G}$ for any of the C-components $\{\mathbf{D}_i\}_{i=1}^k$.*

**Example 3.** Following Example 2, Theorem 3.4 implies that $P_{X_1}(Y_1)$, $P_{X_2}(Y_2)$, $P_{\{X_1, X_2\}}(Y_1)$, and $P_{\{X_1, X_2\}}(Y_2)$ are ID since no Hedge for either $\{Y_1\}$ or $\{Y_2\}$ exists. However, $P_{X_1}(Y_1, Y_2)$ is not

---

[1]Note that in this notation, the order of operations is intervention first, then conditioning.

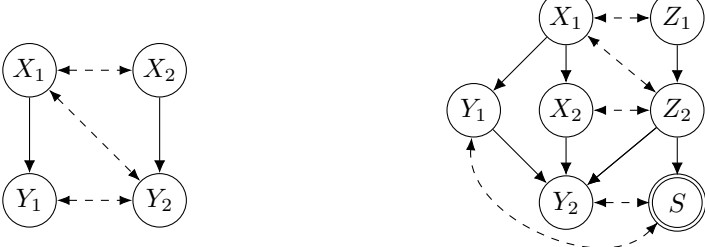

(a) ADMG $\mathcal{G}$ in Examples 2-3.  (b) Augmented ADMG $\mathcal{G}^{\mathrm{s}}$ in Examples 4-9.

Figure 3: ADMGs in Examples of Sections 3, 4, and 5.

ID because $\mathbf{D} = \{Y_1, Y_2, X_2\}$ and the C-components of $Q[\mathbf{D}]$ are $\mathbf{D}_1 = \{Y_1, Y_2\}$ and $\mathbf{D}_2 = \{X_2\}$, and $\{X_1, Y_1, Y_2\}$ (or $\mathbf{V}$) is a Hedge for $\mathbf{D}_1$. Similarly, we can show that $P_{\{X_1, X_2\}}(Y_1, Y_2)$ is not ID.

## 4 S-ID, S-component, and S-Hedge

We begin by providing a formal definition of the S-ID problem in the presence of latent variables, i.e., when the causal graph is an ADMG. Then, we present modifications of the graphical notions from the previous section so that they inherit the key properties of their predecessors and can be applied to the S-ID setting.

To avoid repetition, henceforth, we denote by $\mathbf{V}$ the set of observed variables and by $\mathcal{G}^{\mathrm{s}}$ an augmented ADMG over $\mathbf{V} \cup \{S\}$. Furthermore, we denote by $\mathbf{V}_{\mathrm{AS}}$ and $\mathbf{V}_{\mathrm{NS}}$ the ancestors and non-ancestors of $S$ in $\mathbf{V}$, i.e.,

$$\mathbf{V}_{\mathrm{AS}} := \mathbf{V} \cap Anc_{\mathcal{G}^{\mathrm{s}}}(S), \quad \mathbf{V}_{\mathrm{NS}} := \mathbf{V} \setminus Anc_{\mathcal{G}^{\mathrm{s}}}(S).$$

**Definition 4.1** (S-ID). Let $\mathbf{X}$ and $\mathbf{Y}$ be disjoint subsets of $\mathbf{V}$. Conditional causal effect $P_{\mathbf{X}}(\mathbf{Y}|S = 1)$ (or $P_{\mathbf{X}}^{\mathrm{s}}(\mathbf{Y})$) is S-ID in $\mathcal{G}^{\mathrm{s}}$ if for any two augmented SCMs $\mathcal{M}_1^{\mathrm{s}}$ and $\mathcal{M}_2^{\mathrm{s}}$ with causal graph $\mathcal{G}^{\mathrm{s}}$ for which $P^{\mathcal{M}_1^{\mathrm{s}}}(\mathbf{V}|S = 1) = P^{\mathcal{M}_2^{\mathrm{s}}}(\mathbf{V}|S = 1) > 0$, then $P_{\mathbf{X}}^{\mathcal{M}_1^{\mathrm{s}}}(\mathbf{Y}|S = 1) = P_{\mathbf{X}}^{\mathcal{M}_2^{\mathrm{s}}}(\mathbf{Y}|S = 1)$.

Next definition extends $Q[\cdot]$ and introduces $Q^{\mathrm{s}}[\cdot]$.

**Definition 4.2** ($Q^{\mathrm{s}}[\cdot]$). For $\mathbf{H} \subseteq \mathbf{V}_{\mathrm{NS}}$, we define $Q^{\mathrm{s}}[\mathbf{H}] := P_{\mathbf{V}_{\mathrm{NS}} \setminus \mathbf{H}}\left(\mathbf{H}|Anc_{\mathcal{G}^{\mathrm{s}}}(S) \setminus \{S\}, S = 1\right)$.

The next definition extends C-components (Definition 3.2) and introduces S-components.

**Definition 4.3** (S-component). For a subset $\mathbf{H} \subseteq \mathbf{V}_{\mathrm{NS}}$, let $\mathbf{C}_1, \ldots, \mathbf{C}_k$ denote the C-components of $\mathcal{G}^{\mathrm{s}}[\mathbf{H} \cup Anc_{\mathcal{G}^{\mathrm{s}}}(S)]$. We define the S-components of $\mathbf{H}$ in $\mathcal{G}^{\mathrm{s}}$ as the subsets $\mathbf{H}_i := \mathbf{C}_i \cap \mathbf{H}$ which are non-empty. Furthermore, $\mathbf{H}$ is called a single S-component in $\mathcal{G}^{\mathrm{s}}$ if it contains only one S-component.

Note that $Q^{\mathrm{s}}[\cdot]$ and S-components are only defined for the subsets of $\mathbf{V}_{\mathrm{NS}}$. Figure 4a visualizes the structure of S-components of a subset $\mathbf{H} \subseteq \mathbf{V}_{\mathrm{NS}}$. In this figure, each blue subset (e.g., $M_1$) represents a c-component, which means all the nodes within them are connected via bidirected edges. Therefore, according to Definition 4.3, all nodes inside S-components (e.g., $\mathbf{H}_1$) of $\mathbf{H}$ are connected via bidirected edges in $\mathcal{G}^{\mathrm{s}}[\mathbf{H} \cup \mathbf{V}_{\mathrm{AS}}]$. Figure 4b shows the structure of a single S-component, where all the nodes of $\mathbf{H}$ are connected via bidirected edges in $\mathcal{G}^{\mathrm{s}}[\mathbf{H} \cup \mathbf{V}_{\mathrm{AS}}]$.

**Example 4.** Consider the ADMG $\mathcal{G}^{\mathrm{s}}$ in Figure 3b over $\mathbf{V} \cup \{S\}$, where $\mathbf{V} = \{X_1, X_2, Y_1, Y_2, Z_1, Z_2\}$. Since $Anc_{\mathcal{G}^{\mathrm{s}}}(S) = \{Z_1, Z_2, S\}$, we have $\mathbf{V}_{\mathrm{AS}} = \{Z_1, Z_2\}$ and $\mathbf{V}_{\mathrm{NS}} = \{X_1, X_2, Y_1, Y_2\}$. In this case, the S-components of $\mathbf{V}_{\mathrm{NS}}$ are $\{X_1, X_2\}$ and $\{Y_1, Y_2\}$. Moreover, the S-components of $\{X_1, Y_1, Y_2\}$ are $\{X_1\}$ and $\{Y_1, Y_2\}$.

We now provide two crucial properties for $Q^{\mathrm{s}}[\cdot]$.

**Lemma 4.4.** Let $\mathbf{W}, \mathbf{W}'$ be two subsets of $\mathbf{V}_{\mathrm{NS}}$ such that $\mathbf{W}' \subset \mathbf{W}$. If $\mathbf{W}'$ is an ancestral set in $\mathcal{G}^{\mathrm{s}}[\mathbf{W}]$, then

$$Q^{\mathrm{s}}[\mathbf{W}'] = \sum_{\mathbf{W} \setminus \mathbf{W}'} Q^{\mathrm{s}}[\mathbf{W}]. \tag{1}$$

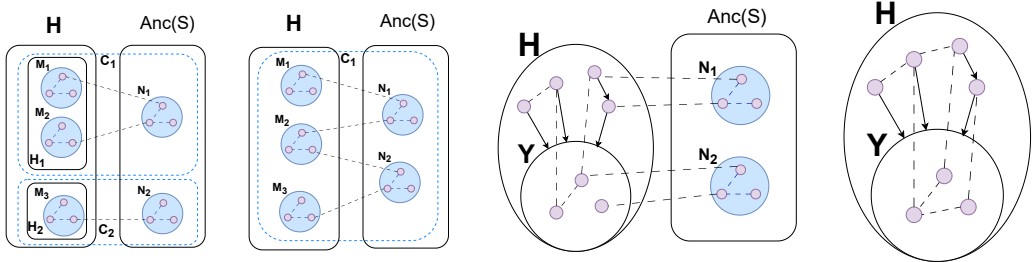

(a) s-components of $\mathbf{H}$ are $\mathbf{H}_1, \mathbf{H}_2$.
(b) $\mathbf{H}$ is a single s-component.
(c) $\mathbf{H}$ is an s-Hedge for $\mathbf{Y}$.
(d) $\mathbf{H}$ is a Hedge for $\mathbf{Y}$.

Figure 4: Visualization of the graph structures defined in Sections 3 and 4.

**Lemma 4.5.** *Suppose $\mathbf{H} \subseteq \mathbf{V}_{\mathrm{NS}}$ and let $\mathbf{H}_1, \ldots, \mathbf{H}_k$ denote the s-components of $\mathbf{H}$ in $\mathcal{G}^{\mathrm{s}}$. Then,*

- *$Q^{\mathrm{s}}[\mathbf{H}]$ decomposes as*

$$Q^{\mathrm{s}}[\mathbf{H}] = Q^{\mathrm{s}}[\mathbf{H}_1] Q^{\mathrm{s}}[\mathbf{H}_2] \ldots Q^{\mathrm{s}}[\mathbf{H}_k]. \tag{2}$$

- *Let $m$ be the number of variables in $\mathbf{H}$, and consider a topological ordering of the variables in graph $\mathcal{G}^{\mathrm{s}}[\mathbf{H}]$, denoted as $V_{h_1} < \cdots < V_{h_m}$. Let $\mathbf{H}^{(0)} = \emptyset$ and for each $1 \leq i \leq m$, $\mathbf{H}^{(i)}$ denote the set of variables in $\mathbf{H}$ ordered before $V_{h_i}$ (including $V_{h_i}$). For every $1 \leq j \leq k$, $Q^{\mathrm{s}}[\mathbf{H}_j]$ can be computed from $Q^{\mathrm{s}}[\mathbf{H}]$ by*

$$Q^{\mathrm{s}}[\mathbf{H}_j] = \prod_{\{i | V_{h_i} \in \mathbf{H}_j\}} \frac{Q^{\mathrm{s}}[\mathbf{H}^{(i)}]}{Q^{\mathrm{s}}[\mathbf{H}^{(i-1)}]}, \tag{3}$$

*where $Q^{\mathrm{s}}[\mathbf{H}^{(i)}]$s can be computed by*

$$Q^{\mathrm{s}}[\mathbf{H}^{(i)}] = \sum_{\mathbf{H} \setminus \mathbf{H}^{(i)}} Q^{\mathrm{s}}[\mathbf{H}]. \tag{4}$$

The aforementioned lemmas are extensions of similar lemmas for $Q[\cdot]$ [TP03] to $Q^{\mathrm{s}}[\cdot]$.

**Example 5.** Following Example 4, since $\{Y_1\}$ is ancestral in $\mathcal{G}^{\mathrm{s}}[Y_1, Y_2]$, Lemma 4.4 implies that $Q^{\mathrm{s}}[Y_1] = \sum_{Y_2} Q^{\mathrm{s}}[Y_1, Y_2]$. Furthermore, since the s-components of $\mathbf{V}_{\mathrm{NS}}$ are $\{X_1, X_2\}$ and $\{Y_1, Y_2\}$, Lemma 4.5 implies that $Q^{\mathrm{s}}[Y_1, Y_2] = \frac{Q^{\mathrm{s}}[\mathbf{V}_{\mathrm{NS}}]}{\sum_{Y_1, Y_2} Q^{\mathrm{s}}[\mathbf{V}_{\mathrm{NS}}]}$. Thus, $Q^{\mathrm{s}}[Y_1]$ can be computed from $Q^{\mathrm{s}}[\mathbf{V}_{\mathrm{NS}}]$.

Finally, we define s-Hedges, which extends Definition 3.3 for Hedges.

**Definition 4.6** (s-Hedge). Suppose $\mathbf{Y} \subseteq \mathbf{V}_{\mathrm{NS}}$ is a single s-component in $\mathcal{G}^{\mathrm{s}}$. A subset $\mathbf{H} \subseteq \mathbf{V}_{\mathrm{NS}}$ is called an s-Hedge for $\mathbf{Y}$ in $\mathcal{G}^{\mathrm{s}}$, if $\mathbf{Y} \subsetneq \mathbf{H}$, $\mathbf{H}$ is a single s-component in $\mathcal{G}^{\mathrm{s}}$, and $\mathbf{H} = Anc_{\mathcal{G}^{\mathrm{s}}[\mathbf{H}]}(\mathbf{Y})$.

When $\mathbf{H} \subseteq \mathbf{V}_{\mathrm{NS}}$ is a single c-component, it is also a single s-component. Therefore, if $\mathbf{H}$ is a Hedge for $\mathbf{Y}$, it will also be an s-Hedge for $\mathbf{Y}$. Thus, Hedges can be seen as special cases of s-Hedges when $\mathbf{H} \subseteq \mathbf{V}_{\mathrm{NS}}$. Figure 4d shows the structure of $\mathbf{H}$, which is a single c-component and forms a Hedge for $\mathbf{Y}$. Moreover, Figure 4c presents the structure of an s-hedge $\mathbf{H}$ for $\mathbf{Y}$. Note that s-hedges are more complex graph structures compared to Hedges. This complexity is required for us to be able to determine whether a causal effect is s-ID.

**Example 6.** Following Examples 4 and 5, $\{X_1, X_2\}$ is an s-Hedge for $\{X_2\}$, because both $\{X_1, X_2\}$ and $\{X_2\}$ are single s-components and $\{X_1, X_2\} = Anc_{\mathcal{G}^{\mathrm{s}}[X_1, X_2]}(X_2)$. Similarly, $\{Y_1, Y_2\}$ is an s-Hedge for $\{Y_2\}$.

## 5 Main Results

In this section, we provide a sufficient graphical condition for a causal effect to be s-ID in an ADMG. This extends the condition presented in [AMK24], which assumes that the causal graph is a DAG. Accordingly, we propose a sound algorithm for the s-ID problem in the presence of latent variables.

Recall that $\mathcal{G}^{\mathrm{s}}$ is an augmented ADMG over the set observed variables $\mathbf{V}$ and auxiliary variable $S$, and we defined $\mathbf{V}_{\mathrm{AS}} = \mathbf{V} \cap Anc_{\mathcal{G}^{\mathrm{s}}}(S)$ and $\mathbf{V}_{\mathrm{NS}} = \mathbf{V} \setminus Anc_{\mathcal{G}^{\mathrm{s}}}(S)$.

**Theorem 5.1.** *For disjoint subsets $\mathbf{X}$ and $\mathbf{Y}$ of $\mathbf{V}$, let $\mathbf{X}_{\mathrm{AS}} := \mathbf{X} \cap \mathbf{V}_{\mathrm{AS}}$, $\mathbf{X}_{\mathrm{NS}} := \mathbf{X} \cap \mathbf{V}_{\mathrm{NS}}$, and $\mathbf{Y}_{\mathrm{NS}} := \mathbf{Y} \cap \mathbf{V}_{\mathrm{NS}}$.*

1. *Conditional causal effect $P_{\mathbf{X}}^{\mathrm{s}}(\mathbf{Y})$ is* s-*ID in $\mathcal{G}^{\mathrm{s}}$ if and only if*

$$(\mathbf{X}_{\mathrm{AS}} \perp\!\!\!\perp \mathbf{Y} | \mathbf{X}_{\mathrm{NS}}, S)_{\mathcal{G}^{\mathrm{s}}_{\underline{\mathbf{X}_{\mathrm{AS}}} \overline{\mathbf{X}_{\mathrm{NS}}}}}, \tag{5}$$

   *and $P_{\mathbf{X}_{\mathrm{NS}}}^{\mathrm{s}}(\mathbf{Y}, \mathbf{X}_{\mathrm{AS}})$ is* s-*ID in $\mathcal{G}^{\mathrm{s}}$.*

2. *Suppose $\mathbf{D} := Anc_{\mathcal{G}^{\mathrm{s}}[\mathbf{V}_{\mathrm{NS}} \setminus \mathbf{X}_{\mathrm{NS}}]}(\mathbf{Y}_{\mathrm{NS}})$ and let $\{\mathbf{D}_i\}_{i=1}^k$ denote the* s-*components of $\mathbf{D}$ in $\mathcal{G}^{\mathrm{s}}$. Conditional causal effect $P_{\mathbf{X}_{\mathrm{NS}}}^{\mathrm{s}}(\mathbf{Y}, \mathbf{X}_{\mathrm{AS}})$ is* s-*ID in $\mathcal{G}^{\mathrm{s}}$ if there are no* s-*Hedge in $\mathcal{G}^{\mathrm{s}}$ for any of $\{\mathbf{D}_i\}_{i=1}^k$.*

**Remark 5.2.** If either $\mathbf{X}_{\mathrm{NS}}$ or $\mathbf{Y}_{\mathrm{NS}}$ is an empty set, then $P_{\mathbf{X}}^{\mathrm{s}}(\mathbf{Y})$ is s-ID in $\mathcal{G}^{\mathrm{s}}$ if and only if Equation (5) holds.

In the absence of latent variables, i.e., when $\mathcal{G}^{\mathrm{s}}$ is a directed acyclic graph (DAG), there are no s-Hedge in $\mathcal{G}^{\mathrm{s}}$ since all the edges are directed. Therefore, Theorem 5.1 states that $P_{\mathbf{X}_{\mathrm{NS}}}^{\mathrm{s}}(\mathbf{Y}, \mathbf{X}_{\mathrm{AS}})$ is always s-ID, and $P_{\mathbf{X}}^{\mathrm{s}}(\mathbf{Y})$ is s-ID in $\mathcal{G}^{\mathrm{s}}$ if and only if Equation (5) holds. We note that this is consistent with the condition presented in [AMK24, Theorem 2] for the s-ID problem in the absence of latent variables.

**Example 7.** Consider again ADMG $\mathcal{G}^{\mathrm{s}}$ in Figure 3b, where we want to determine whether $P_{\{X_1, X_2, Z_1\}}^{\mathrm{s}}(Y_1, Y_2)$ is s-ID in $\mathcal{G}^{\mathrm{s}}$. In this case, $\mathbf{X}_{\mathrm{AS}} = \{Z_1\}$, $\mathbf{X}_{\mathrm{NS}} = \{X_1, X_2\}$, $\mathbf{Y}_{\mathrm{NS}} = \{Y_1, Y_2\}$, and $(Z_1 \perp\!\!\!\perp \{Y_1, Y_2\} | X_1, X_2, S)_{\mathcal{G}^{\mathrm{s}}_{\underline{Z_1} \overline{X_1, X_2}}}$. This shows that Equation (5) holds. Hence, we need to determine whether $P_{X_1, X_2}^{\mathrm{s}}(Y_1, Y_2, Z_1)$ is s-ID in $\mathcal{G}^{\mathrm{s}}$. In this case, $\mathbf{D} = Anc_{\mathcal{G}^{\mathrm{s}}[Y_1, Y_2]}(Y_1, Y_2) = \{Y_1, Y_2\}$, which is a single s-component. Since there exists no s-Hedge for $\{Y_1, Y_2\}$, Theorem 5.1 implies that $P_{\{X_1, X_2, Z_1\}}^{\mathrm{s}}(Y_1, Y_2)$ is s-ID in $\mathcal{G}^{\mathrm{s}}$.

**Algorithm for s-ID.** So far, we have presented a graphical condition to determine whether a causal effect is s-ID. In this section, we propose a recursive algorithm that returns an expression for $P_{\mathbf{X}}^{\mathrm{s}}(\mathbf{Y})$ in terms of $P^{\mathrm{s}}(\mathbf{V})$ when the condition of Theorem 5.1 holds, and otherwise, returns FAIL.

In the proof of Theorem 5.1 presented in the appendix, we show that when Equation (5) holds, then

$$P_{\mathbf{X}}^{\mathrm{s}}(\mathbf{Y}) = \sum_{\mathbf{W}} P^{\mathrm{s}}(\mathbf{Y}_{\mathrm{AS}}, \mathbf{W} | \mathbf{X}_{\mathrm{AS}}) P_{\mathbf{X}_{\mathrm{NS}}}^{\mathrm{s}}(\mathbf{Y}_{\mathrm{NS}} | \mathbf{V}_{\mathrm{AS}}), \tag{6}$$

where $\mathbf{W} = \mathbf{V}_{\mathrm{AS}} \setminus (\mathbf{X}_{\mathrm{AS}} \cup \mathbf{Y}_{\mathrm{AS}})$. Thus, it suffices to find an expression for $P_{\mathbf{X}_{\mathrm{NS}}}^{\mathrm{s}}(\mathbf{Y}_{\mathrm{NS}} | \mathbf{V}_{\mathrm{AS}})$ in terms of $P^{\mathrm{s}}(\mathbf{V})$. Let $\mathbf{D} = Anc_{\mathcal{G}^{\mathrm{s}}[\mathbf{V}_{\mathrm{NS}} \setminus \mathbf{X}_{\mathrm{NS}}]}(\mathbf{Y}_{\mathrm{NS}})$ and $\{\mathbf{D}_i\}_{i=1}^k$ be the s-components of $\mathbf{D}$ in $\mathcal{G}^{\mathrm{s}}$. From Lemmas 4.4 and 4.5 we have

$$P_{\mathbf{X}_{\mathrm{NS}}}^{\mathrm{s}}(\mathbf{Y}_{\mathrm{NS}} | \mathbf{V}_{\mathrm{AS}}) = \sum_{\mathbf{D} \setminus \mathbf{Y}_{\mathrm{NS}}} \prod_i Q^{\mathrm{s}}[\mathbf{D}_i]. \tag{7}$$

Therefore, it suffices to find an expression for each $\mathbf{D}_i$ in terms of $P^{\mathrm{s}}(\mathbf{V})$. Note that $\mathbf{D}_i$ is a single s-component in $\mathcal{G}^{\mathrm{s}}$. We can now propose Algorithm 1 for computing $P_{\mathbf{X}}^{\mathrm{s}}(\mathbf{Y})$ from $P^{\mathrm{s}}(\mathbf{V})$.

Function **sID** takes disjoint subsets $\mathbf{X}$ and $\mathbf{Y}$ of $\mathbf{V}$ along with an augmented ADMG $\mathcal{G}^{\mathrm{s}}$ and conditional distribution $P^{\mathrm{s}}(\mathbf{V})$ as input. After defining the required notations in lines 3-5, it checks Equation (5) in line 6. If this condition is met, it defines $\mathbf{D}$ and its s-components $\{\mathbf{D}_i\}_{i=1}^k$ in $\mathcal{G}^{\mathrm{s}}$. For each $1 \leq i \leq k$, it finds the corresponding s-component $\mathbf{T}_i$ of $\mathbf{V}_{\mathrm{NS}}$ in $\mathcal{G}^{\mathrm{s}}$ that contains $\mathbf{D}_i$. Note that $\mathbf{T}_i$ is well-defined as $\mathbf{D}_i$ cannot partially intersect with the s-components of $\mathbf{V}_{\mathrm{NS}}$ in $\mathcal{G}^{\mathrm{s}}$. Next, the algorithm seeks to compute $Q^{\mathrm{s}}[\mathbf{D}_i]$ from $Q^{\mathrm{s}}[\mathbf{T}_i]$ by calling Function **sID-Single**. If Function **sID-Single** succeeds in returning an expression for each $i$, then the algorithm uses Equations (6) and (7) to return an expression for $P_{\mathbf{X}}^{\mathrm{s}}(\mathbf{Y})$ in terms of $P^{\mathrm{s}}(\mathbf{Y})$. Otherwise, the algorithm returns FAIL.

Function **sID-Single** takes two single s-components $\mathbf{C}$ and $\mathbf{T}$ in $\mathcal{G}^{\mathrm{s}}$ such that $\mathbf{C} \subseteq \mathbf{T}$ and aims to drive an expression for $Q^{\mathrm{s}}[\mathbf{C}]$ in terms of $Q^{\mathrm{s}}[\mathbf{T}]$. The procedure is recursive and uses Lemmas 4.4 and 4.5. In each recursion, the algorithm reduces $\mathbf{T}$ to a smaller subset $\mathbf{T}'$ such that $\mathbf{T}'$ is still a single s-component in $\mathcal{G}^{\mathrm{s}}$ and $\mathbf{C} \subseteq \mathbf{T}'$ (lines 7-10). Eventually, the function either returns FAIL or an expression for $Q^{\mathrm{s}}[\mathbf{C}]$.

**Algorithm 1** Computing $P_\mathbf{X}^\mathrm{s}(\mathbf{Y})$ from $P^\mathrm{s}(\mathbf{V})$

1: **Function sID($\mathbf{X}, \mathbf{Y}, \mathcal{G}^\mathrm{s}, P^\mathrm{s}(\mathbf{V})$)**
2: **Output:** Expression for $P_\mathbf{X}^\mathrm{s}(\mathbf{Y})$ in terms of $P^\mathrm{s}$ or FAIL
3: $\mathbf{V}_\mathrm{AS} \leftarrow \mathbf{V} \cap Anc_{\mathcal{G}^\mathrm{s}}(S),\ \mathbf{V}_\mathrm{NS} \leftarrow \mathbf{V} \setminus Anc_{\mathcal{G}^\mathrm{s}}(S)$
4: $\mathbf{X}_\mathrm{AS} \leftarrow \mathbf{X} \cap \mathbf{V}_\mathrm{AS},\quad \mathbf{X}_\mathrm{NS} \leftarrow \mathbf{X} \cap \mathbf{V}_\mathrm{NS}$
5: $\mathbf{Y}_\mathrm{AS} \leftarrow \mathbf{Y} \cap \mathbf{V}_\mathrm{AS},\quad \mathbf{Y}_\mathrm{NS} \leftarrow \mathbf{Y} \cap \mathbf{V}_\mathrm{NS}$
6: **if** $(\mathbf{X}_\mathrm{AS} \not\perp\!\!\!\perp \mathbf{Y} | \mathbf{X}_\mathrm{NS}, S)_{\mathcal{G}^\mathrm{s}_{\underline{\mathbf{X}_\mathrm{AS}}\overline{\mathbf{X}_\mathrm{NS}}}}$ **then**
7:    **Return** FAIL
8: $\mathbf{D} \leftarrow Anc_{\mathcal{G}^\mathrm{s}[\mathbf{V}_\mathrm{NS} \setminus \mathbf{X}_\mathrm{NS}]}(\mathbf{Y}_\mathrm{NS})$
9: $\{\mathbf{D}_1, \ldots, \mathbf{D}_k\} \leftarrow$ S-components of $\mathbf{D}$ in $\mathcal{G}^\mathrm{s}$
10: **for** $i$ in $[1:k]$ **do**
11:    $\mathbf{T}_i \leftarrow$ The S-component of $\mathbf{V}_\mathrm{NS}$ that contains $\mathbf{D}_i$
12:    Compute $Q^\mathrm{s}[\mathbf{T}_i]$ using Lemma 4.5
13:    $Q^\mathrm{s}[\mathbf{D}_i] \leftarrow$ **sID-Single**($\mathbf{D}_i, \mathbf{T}_i, Q^\mathrm{s}[\mathbf{T}_i]$)
14:    **if** $Q^\mathrm{s}[\mathbf{D}_i] =$ FAIL **then**
15:       **Return** FAIL
16: $\mathbf{W} \leftarrow \mathbf{V}_\mathrm{AS} \setminus (\mathbf{X}_\mathrm{AS} \cup \mathbf{Y}_\mathrm{AS})$
17: **Return** $\sum_\mathbf{W} P^\mathrm{s}(\mathbf{Y}_\mathrm{AS}, \mathbf{W} | \mathbf{X}_\mathrm{AS}) \sum_{\mathbf{D} \setminus \mathbf{Y}_\mathrm{NS}} \prod_i Q^\mathrm{s}[\mathbf{D}_i]$

1: **Function sID-Single($\mathbf{C}, \mathbf{T}, Q^\mathrm{s}[\mathbf{T}]$)**
2: **Input**: Two single s-components $\mathbf{C}$ and $\mathbf{T}$ in $\mathcal{G}^\mathrm{s}$ such that $\mathbf{C} \subseteq \mathbf{T}$
3: **Output:** Expression for $Q^\mathrm{s}[\mathbf{C}]$ in terms of $Q^\mathrm{s}[\mathbf{T}]$ or FAIL
4: $\mathbf{A} \leftarrow Anc_{\mathcal{G}^\mathrm{s}[\mathbf{T}]}(\mathbf{C})$
5: **if** $\mathbf{A} = \mathbf{C}$: **Return** $\sum_{\mathbf{T} \setminus \mathbf{C}} Q^\mathrm{s}[\mathbf{T}]$
6: **if** $\mathbf{A} = \mathbf{T}$: **Return** FAIL
7: **if** $\mathbf{C} \subsetneq \mathbf{A} \subsetneq \mathbf{T}$, **then**
8:    $\mathbf{T}' \leftarrow$ The s-component of $\mathbf{A}$ in $\mathcal{G}^\mathrm{s}$ that contains $\mathbf{C}$
9:    Compute $Q^\mathrm{s}[\mathbf{T}']$ from $Q^\mathrm{s}[\mathbf{T}]$ using Lemma 4.5
10:    **Return sID-Single**($\mathbf{C}, \mathbf{T}', Q^\mathrm{s}[\mathbf{T}']$)

**Example 8.** Consider again ADMG $\mathcal{G}^\mathrm{s}$ depicted in Figure 3b, where we want to apply Algorithm 1 for causal effect $P_{X_2}^\mathrm{s}(Y_2)$. Herein, $\mathbf{X}_\mathrm{NS} = \{X_2\}$, $\mathbf{Y}_\mathrm{NS} = \{Y_2\}$, and $\mathbf{X}_\mathrm{AS} = \mathbf{Y}_\mathrm{AS} = \varnothing$. Function **S-ID** passes the condition in line 6 and defines $\mathbf{D} = \{X_1, Y_1, Y_2\}$, leading to $\mathbf{D}_1 = \{X_1\}$ and $\mathbf{D}_2 = \{Y_1, Y_2\}$. It then defines $\mathbf{T}_i$'s in line 12 as $\mathbf{T}_1 = \{X_1, X_2\}$ and $\mathbf{T}_2 = \{Y_1, Y_2\}$. In line 13, it uses Lemma 4.5 to compute $Q^\mathrm{s}[\mathbf{T}_1] = \sum_{Y_1, Y_2} Q^\mathrm{s}[\mathbf{V}_\mathrm{NS}]$ and $Q^\mathrm{s}[\mathbf{T}_2] = \frac{Q^\mathrm{s}[\mathbf{V}_\mathrm{NS}]}{\sum_{Y_1, Y_2} Q^\mathrm{s}[\mathbf{V}_\mathrm{NS}]}$ (See Example 5). It then calls Function **sID-Single**, which returns $Q^\mathrm{s}[\mathbf{D}_1] = \sum_{X_2} Q^\mathrm{s}[\mathbf{T}_1]$ and $Q^\mathrm{s}[\mathbf{D}_2] = Q^\mathrm{s}[\mathbf{T}_2]$. Finally, in line 20, the function returns

$$P_{X_2}^\mathrm{s}(Y_2) = \sum_{Z_1, Z_2} P^\mathrm{s}(Z_1, Z_2) \sum_{X_1, Y_1} Q^\mathrm{s}[X_1] Q^\mathrm{s}[Y_1, Y_2],$$

where $Q^\mathrm{s}[X_1] = P^\mathrm{s}(X_1 | Z_1, Z_2)$ and $Q^\mathrm{s}[Y_1, Y_2] = P^\mathrm{s}(Y_1, Y_2 | X_1, X_2, Z_1, Z_2)$.

**Example 9.** Following the previous example, suppose we want to apply Algorithm 1 for computing causal effect $P_{X_1}^\mathrm{s}(Y_1, Y_2)$. In this case, the algorithm needs to compute $Q^\mathrm{s}[Y_1, Y_2]$ and $Q^\mathrm{s}[X_2]$. However, when the algorithm calls **sID-Single** $(X_2, \{X_1, X_2\}, Q^\mathrm{s}[X_1, X_2])$, Function **sID-Single** returns FAIL. Accordingly, Function **sID** returns FAIL for $P_{X_1}^\mathrm{s}(Y_1, Y_2)$.

**Remark 5.3.** Algorithm 1 is sound for the S-ID problem in the presence of latent variables. We conjecture that this algorithm is also *complete*, meaning that whenever it returns FAIL, the corresponding causal effect is not S-ID.

## 6 Reduction from S-Recoverability to S-ID

Recall that the objective in S-Recoverability is to compute $P_\mathbf{X}(\mathbf{Y})$ from $P^\mathrm{s}(\mathbf{V})$ [BTP14], while S-ID aims to compute $P_\mathbf{X}^\mathrm{s}(\mathbf{Y})$ from $P^\mathrm{s}(\mathbf{V})$. [BT15] proposed RC, a sound algorithm for the S-Recoverability problem. Subsequently, [CTB19] proved that RC is complete. In this section, we present a reduction from the S-Recoverability problem to the S-ID problem. This indicates that solving S-ID can solve the S-Recoverability problem (but not the other way around).

**Theorem 6.1.** *For disjoint subsets $\mathbf{X}$ and $\mathbf{Y}$ of $\mathbf{V}$, $P_\mathbf{X}(\mathbf{Y})$ can be uniquely computed from $P^\mathrm{s}(\mathbf{V})$ in the augmented ADMG $\mathcal{G}^\mathrm{s}$ if and only if*

$$(\mathbf{Y} \perp\!\!\!\perp S | \mathbf{X})_{\mathcal{G}^\mathrm{s}_{\overline{\mathbf{X}}}}, \tag{8}$$

*and $P_\mathbf{X}^\mathrm{s}(\mathbf{Y})$ is S-ID in $\mathcal{G}^\mathrm{s}$.*

**Algorithm 2**
Reduction from S-Recoverability to S-ID

1: **Input:** $\mathbf{X}, \mathbf{Y}, \mathcal{G}^{\mathrm{s}}, P^{\mathrm{s}}$
2: **Output:** Expression for $P_{\mathbf{X}}(\mathbf{Y})$ in terms of $P^{\mathrm{s}}$ or FAIL
3: **if** $(\mathbf{Y} \not\perp\!\!\!\perp S | \mathbf{X})_{\mathcal{G}^{\mathrm{s}}_{\overline{\mathbf{X}}}}$ **then**
4:     **Return** FAIL
5: **else**
6:     **Return** $\mathbf{sID}(\mathbf{X}, \mathbf{Y}, \mathcal{G}^{\mathrm{s}}, P^{\mathrm{s}})$

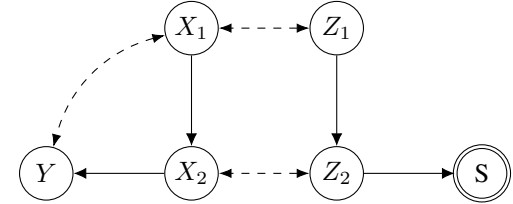

Figure 5: The augmented ADMG $\mathcal{G}^{\mathrm{s}}$ in Example 10.

**Remark 6.2.** Equation (8) is a very restrictive condition, and when it holds, Rule 1 of do-calculus implies that $P_{\mathbf{X}}(\mathbf{Y}) = P_{\mathbf{X}}^{\mathrm{s}}(\mathbf{Y})$.

Theorem 6.1 implies that when Equation (8) does not hold, then $P_{\mathbf{X}}(\mathbf{Y})$ is not S-Recoverable. However, this causal effect might be identifiable in the target sub-population, i.e., $P_{\mathbf{X}}^{\mathrm{s}}(\mathbf{Y})$ is S-ID.

As a consequence of Theorem 6.1, we propose Algorithm 2 for computing $P_{\mathbf{X}}(\mathbf{Y})$ from $P^{\mathrm{s}}(\mathbf{V})$. The algorithm takes as input two disjoint subsets $\mathbf{X}$ and $\mathbf{Y}$ of $\mathbf{V}$ along with an augmented ADMG over $\mathbf{V} \cup \{S\}$ and the conditional distribution $P^{\mathrm{s}}(\mathbf{V})$. It first checks Equation (8) in line 3, and then calls Algorithm 1 as a subroutine to compute $P_{\mathbf{X}}^{\mathrm{s}}(\mathbf{Y})$ from $P^{\mathrm{s}}(\mathbf{V})$ when it is S-ID in $\mathcal{G}^{\mathrm{s}}$.

**Example 10.** Consider the augmented ADMG $\mathcal{G}^{\mathrm{s}}$ in Figure 5. In this graph, $(Y \not\perp\!\!\!\perp S | X_1)_{\mathcal{G}^{\mathrm{s}}_{\overline{X_1}}}$, thus, Theorem 6.1 implies that $P_{X_1}(Y)$ cannot be uniquely computed from $P^{\mathrm{s}}(\mathbf{V})$. On the other hand, $P_{X_2}(Y)$ can be identified from $P^{\mathrm{s}}(\mathbf{V})$ since $(Y \perp\!\!\!\perp S | X_2)_{\mathcal{G}^{\mathrm{s}}_{\overline{X_2}}}$ and due to Theorem 5.1, $P_{X_2}^{\mathrm{s}}(Y)$ is S-ID in $\mathcal{G}^{\mathrm{s}}$. In this case, Algorithm 2 returns the following expression for $P_{X_2}(Y)$ in terms of $P^{\mathrm{s}}$

$$P_{X_2}(Y) = \sum_{Z_1, Z_2} P^{\mathrm{s}}(Z_1, Z_2) \sum_{X_1} \frac{P^{\mathrm{s}}(X_1, X_2, Y | Z_1, Z_2)}{P^{\mathrm{s}}(X_2 | X_1, Z_1, Z_2)}.$$

# 7 Conclusion

The S-ID problem, introduced by [AMK24], asks whether, given the causal graph, a causal effect in a sub-population can be identified from the observational distribution pertaining to the same sub-population. [AMK24] addressed this problem when all the variables in the causal graph are observable. In this paper, we studied the S-ID problem in the presence of latent variables and provided a sufficient graphical condition to determine whether a causal effect is S-ID. Consequently, we proposed a sound algorithm for S-ID. While this paper proves the soundness of our proposed method, we also conjecture that our approach is not only sound but also complete. Finally, by presenting an appropriate reduction, we showed that solving S-ID can solve the S-Recoverability problem.

## Acknowledgments

We thank the anonymous reviewers for their feedback. This research was in part supported by the Swiss National Science Foundation under NCCR Automation, grant agreement 51NF40_180545 and Swiss SNF project 200021_204355 /1.

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

# Appendix

The structure of the appendix is as follows. Appendix A includes an additional example of S-ID problem in the presence of latent variables. In Appendix B, we provide some preliminary lemmas used throughout our proofs. The proofs for the main results, namely, Lemmas 4.4, 4.5, and Theorems 5.1, 6.1 are presented in Appendix C. In Appendix D, we will conduct an experiment to compare the outputs of S-ID algorithm and the classic ID algorithm.

## A  Additional Example

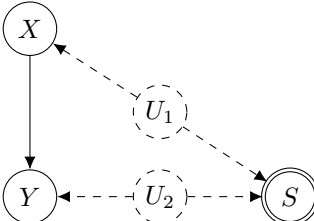

Figure 6: ADMG $\mathcal{G}^S$ of the example of Appendix A.

In this section, we provide an example where ignoring $S$ results in an identifiable effect, whereas the target causal effect is, in fact, not S-ID.

Consider the following two SCMs.

SCM $\mathcal{M}_1$:

$$U_1 \sim Bern(0.5)$$
$$U_2 \sim Bern(0.5)$$
$$\varepsilon_y \sim Bern(0.3)$$
$$X = U_1$$
$$Y = X \oplus U_2 \oplus \varepsilon_y$$
$$S = \overline{U_1 \oplus U_2}$$

where $\oplus$ denotes the XOR operator, and $Bern(p)$ denotes a Bernoulli random variable with parameter $p$.

SCM $\mathcal{M}_2$:

$$U_1 \sim Bern(0.5)$$
$$U_2 \sim Bern(0.5)$$
$$\varepsilon_y \sim Bern(0.3)$$
$$X = U_1$$
$$Y = \varepsilon_y$$
$$S = 1$$

According to the above equations, we have

$$P^{\mathcal{M}_1}(X = x, Y = y | S = 1) = P(U_1 = x)P(\varepsilon_y = y) = 0.5 \times P(\varepsilon_y = y) > 0.$$

Similarly for $\mathcal{M}_2$ we have

$$P^{\mathcal{M}_2}(X = x, Y = y | S = 1) = P^{\mathcal{M}_2}(X = x, Y = y)$$
$$= P^{\mathcal{M}_2}(X = x)P^{\mathcal{M}_2}(Y = y)$$
$$= P(U_1 = x)P(\varepsilon_y = y) = 0.5 \times P(\varepsilon_y = y) > 0.$$

Thus, $P^{\mathcal{M}_1}(X, Y | S = 1) = P^{\mathcal{M}_2}(X, Y | S = 1) > 0.$ Furthermore, we have

$$P^{\mathcal{M}_1}_{x=0}(Y = 1 | S = 1) = P^{\mathcal{M}_1}_{x=0}(U_2 + \varepsilon_y = 1 | S = 1) = P^{\mathcal{M}_1}(U_2 + \varepsilon_y = 1) = 0.5.$$

Similarly, for SCM $\mathcal{M}_2$:

$$P_{x=0}^{\mathcal{M}_2}(Y=1|S=1) = P^{\mathcal{M}_2}(\varepsilon_y = 1|S=1) = P(\varepsilon_y = 1) = 0.3$$

This shows that $P_X^{\mathrm{s}}(Y)$ is not s-ID in this causal graph, as $P^{\mathcal{M}_1}(X, Y|S=1) = P^{\mathcal{M}_2}(X, Y|S=1) > 0$, but $P_{x=0}^{\mathcal{M}_1}(Y=1|S=1) \neq P_{x=0}^{\mathcal{M}_2}(Y=1|S=1)$.

Note that ignoring the sub-population, the causal effect $P_X(Y)$ is clearly identifiable from $P(X, Y)$, but as we showed above, the causal effect of $X$ on $Y$ is not identifiable in the sub-population from the observational data of that sub-population.

## B  Technical Preliminaries

**Pearl's do-calculus rules [Pea00]**: Let $\mathbf{X}, \mathbf{Y}, \mathbf{Z}, \mathbf{W}$ be four disjoint subsets of $\mathbf{V}$. The following three rules, commonly referred to as Pearl's do-calculus rules [Pea00], provide a tool for calculating interventional distributions using the causal graph.

- **Rule 1**: If $(\mathbf{Y} \perp\!\!\!\perp \mathbf{Z}|\mathbf{X}, \mathbf{W})_{\mathcal{G}_{\overline{\mathbf{X}}}}$, then

$$P_{\mathbf{X}}(\mathbf{Y}|\mathbf{Z}, \mathbf{W}) = P_{\mathbf{X}}(\mathbf{Y}|\mathbf{W}).$$

- **Rule 2**: If $(\mathbf{Y} \perp\!\!\!\perp \mathbf{Z}|\mathbf{X}, \mathbf{W})_{\mathcal{G}_{\overline{\mathbf{X}}\underline{\mathbf{Z}}}}$, then

$$P_{\mathbf{X}, \mathbf{Z}}(\mathbf{Y}|\mathbf{W}) = P_{\mathbf{X}}(\mathbf{Y}|\mathbf{Z}, \mathbf{W}).$$

- **Rule 3**: If $(\mathbf{Y} \perp\!\!\!\perp \mathbf{Z}|\mathbf{X}, \mathbf{W})_{\mathcal{G}_{\overline{\mathbf{X}\mathbf{Z}(W)}}}$, where $\mathbf{Z}(\mathbf{W}) := \mathbf{Z} \setminus Anc_{\mathcal{G}_{\overline{\mathbf{X}}}}(\mathbf{W})$, then

$$P_{\mathbf{X}, \mathbf{Z}}(\mathbf{Y}|\mathbf{W}) = P_{\mathbf{X}}(\mathbf{Y}|\mathbf{W}).$$

**Lemma B.1** (TP03). *For two sets $\mathbf{W}' \subset \mathbf{W}$, if $\mathbf{W}'$ is an ancestral set in $G[\mathbf{W}]$, then*

$$Q[\mathbf{W}'] = \sum_{\mathbf{W}\setminus\mathbf{W}'} Q[\mathbf{W}]. \tag{9}$$

**Lemma B.2** (TP03). *Let $\mathbf{C} \subseteq \mathbf{V}$, and assume that C is partitioned into C-components $\mathbf{C}_1, \ldots, \mathbf{C}_m$ in the subgraph $G[\mathbf{C}]$. Then we have*

- *$Q[\mathbf{C}]$ decomposes as*

$$Q[\mathbf{C}] = \prod_{i=1}^{m} Q[\mathbf{C}_i] \tag{10}$$

- *Let $k$ denote the number of variables in $\mathbf{H}$, and let us assume a topological order of variables in $\mathbf{C}$ as $V_{c_1} < V_{c_2} < \cdots < V_{c_k}$ in $G_{\mathbf{C}}$. Let $\mathbf{C}^i$ be the set of variables in $\mathbf{C}$ ordered before $V_{c_i}$ (including $V_{c_i}$), for $i = 1, 2, \ldots, k$, where $\mathbf{C}^0$ is an empty set. Then each $Q[\mathbf{C}_j]$, $j = 1, 2, \ldots, m$, is computable from $Q[\mathbf{C}]$ and is given by*

$$Q[\mathbf{C}_j] = \prod_{\{i|V_{c_i}\in\mathbf{C}_j\}} \frac{Q[\mathbf{C}^{(i)}]}{Q[\mathbf{C}^{(i-1)}]}, \tag{11}$$

*where each $Q[\mathbf{C}^i]$ is given by*

$$Q[\mathbf{C}^{(i)}] = \sum_{\mathbf{C}\setminus\mathbf{C}^{(i)}} Q[\mathbf{C}]. \tag{12}$$

**Corollary B.3.** *Let $\mathbf{H}_1 \sqcup \mathbf{H}_2 \sqcup \cdots \sqcup \mathbf{H}_m$ be a partition of set $\mathbf{C}$, where for each $\mathbf{C}_i$ there exists $1 \leq j \leq m$ such that $\mathbf{C}_i \subseteq \mathbf{H}_j$. Then, we have*

$$Q[\mathbf{C}] = \prod_j Q[\mathbf{H}_j].$$

**Lemma B.4.** *Let $\mathbf{H}$ be a subset of $\mathbf{V}_{\mathrm{NS}}$, we have the following equation*

$$Q^{\mathrm{s}}[\mathbf{H}] = \frac{Q[\mathbf{H} \cup Anc(S)]}{Q[Anc(S)]}. \tag{13}$$

*Proof.* According to definition of $Q^{\text{S}}$, we have

$$Q^{\text{S}}[\mathbf{H}] = P_{\mathbf{V}_{\text{NS}}\setminus\mathbf{H}}(\mathbf{H}|Anc(S)) = \frac{P_{\mathbf{V}_{\text{NS}}\setminus\mathbf{H}}(\mathbf{H}, Anc(S))}{P_{\mathbf{V}_{\text{NS}}\setminus\mathbf{H}}(Anc(S))} = \frac{Q[\mathbf{H} \cup Anc(S)]}{P_{\mathbf{V}_{\text{NS}}\setminus\mathbf{H}}(Anc(S))}.$$

Note that $Anc(S)$ is an ancestral set in $\mathcal{G}$; Hence, for any $\mathbf{H}$, we have

$$P_{\mathbf{V}_{\text{NS}}\setminus\mathbf{H}}(Anc(S)) = P(Anc(S)) = Q[Anc(S)].$$

This implies Equation (13). □

**Lemma B.5.** *Let $\mathbf{X}_{\text{NS}}$ and $\mathbf{Y}_{\text{NS}}$ be disjoint subsets of $\mathbf{V}_{\text{NS}} = \mathbf{V} \setminus Anc_{\mathcal{G}^{\text{S}}}(S)$. We have*

$$P_{\mathbf{X}_{\text{NS}}}(\mathbf{Y}_{\text{NS}}|Anc(S)) = \sum_{\mathbf{D}\setminus\mathbf{Y}_{\text{NS}}} Q^{\text{S}}[\mathbf{D}_1]\ldots Q^{\text{S}}[\mathbf{D}_k],$$

*where $\mathbf{D} = Anc_{\mathcal{G}[\mathbf{V}_{\text{NS}}\setminus\mathbf{X}_{\text{NS}}]}(\mathbf{Y}_{\text{NS}})$, and $\mathbf{D}_i$'s are S-components of $\mathbf{D}$.*

*Proof.* Using marginalization over $\mathbf{V}_{\text{NS}} \setminus (\mathbf{X}_{\text{NS}} \cup \mathbf{Y}_{\text{NS}})$, we have

$$P_{\mathbf{X}_{\text{NS}}}(\mathbf{Y}_{\text{NS}}|Anc(S)) = \sum_{\mathbf{V}_{\text{NS}}\setminus(\mathbf{X}_{\text{NS}}\cup\mathbf{Y}_{\text{NS}})} P_{\mathbf{X}_{\text{NS}}}(\mathbf{V}_{\text{NS}} \setminus \mathbf{X}_{\text{NS}}|Anc(S)) = \sum_{\mathbf{V}_{\text{NS}}\setminus(\mathbf{X}_{\text{NS}}\cup\mathbf{Y}_{\text{NS}})} Q^{\text{S}}[\mathbf{V}_{\text{NS}} \setminus \mathbf{X}_{\text{NS}}].$$

Since $\mathbf{D}$ is an ancestral set in $\mathcal{G}[\mathbf{V}_{\text{NS}} \setminus \mathbf{X}_{\text{NS}}]$, according to Lemma 4.4, we have

$$P_{\mathbf{X}_{\text{NS}}}(\mathbf{Y}_{\text{NS}}|Anc(S)) = \sum_{\mathbf{V}_{\text{NS}}\setminus(\mathbf{X}_{\text{NS}}\cup\mathbf{Y}_{\text{NS}})} Q^{\text{S}}[\mathbf{V}_{\text{NS}}\setminus\mathbf{X}_{\text{NS}}] = \sum_{\mathbf{D}\setminus\mathbf{Y}_{\text{NS}}} \sum_{\mathbf{V}_{\text{NS}}\setminus(\mathbf{X}_{\text{NS}}\cup\mathbf{D})} Q^{\text{S}}[\mathbf{V}_{\text{NS}}\setminus\mathbf{X}_{\text{NS}}] = \sum_{\mathbf{D}\setminus\mathbf{Y}_{\text{NS}}} Q[\mathbf{D}].$$

Therefore, the first property of Lemma 4.5 implies that

$$P_{\mathbf{X}_{\text{NS}}}(\mathbf{Y}_{\text{NS}}|Anc(S)) = \sum_{\mathbf{D}\setminus\mathbf{Y}_{\text{NS}}} Q^{\text{S}}[\mathbf{D}_1]\ldots Q^{\text{S}}[\mathbf{D}_k].$$

□

**Lemma B.6.** *If Function sID-Single returns* FAIL *for the inputs $\mathbf{C}$ and $\mathbf{T}$, then $\mathbf{T}$ is an S-Hedge for $\mathbf{C}$.*

*Proof.* When the algorithm returns Fail,

1. $\mathbf{C}$ and $\mathbf{T}$ are both single S-components since the inputs of the algorithm have to be S-components.

2. If $\mathbf{A} := Anc_{\mathcal{G}[\mathbf{T}]}(\mathbf{C})$, then $\mathbf{T} = \mathbf{A}$.

According to the definition of s-Hedge 4.6, $\mathbf{T}$ is an s-Hedge for $\mathbf{C}$. □

## C   Proofs of Main Results

**Lemma 4.4.** *Let $\mathbf{W}, \mathbf{W}'$ be two subsets of $\mathbf{V}_{\text{NS}}$ such that $\mathbf{W}' \subset \mathbf{W}$. If $\mathbf{W}'$ is an ancestral set in $G[\mathbf{W}]$, then we have*

$$Q^{\text{S}}[\mathbf{W}'] = \sum_{\mathbf{W}\setminus\mathbf{W}'} Q^{\text{S}}[\mathbf{W}]. \tag{14}$$

*Proof.* According to Lemma B.4, we have the following equations

$$Q^{\text{S}}[\mathbf{W}] = \frac{Q[\mathbf{W} \cup Anc(S)]}{Q[Anc(S)]},$$

$$Q^{\text{S}}[\mathbf{W}'] = \frac{Q[\mathbf{W}' \cup Anc(S)]}{Q[Anc(S)]}.$$

Therefore, by replacing the above equations in Equation (1), it is sufficient to show that

$$\frac{Q[\mathbf{W}' \cup Anc(S)]}{Q[Anc(S)]} = \sum_{\mathbf{W} \setminus \mathbf{W}'} \frac{Q[\mathbf{W} \cup Anc(S)]}{Q[Anc(S)]}.$$

Since $P(Anc(S)) > 0$, this is equivalent to

$$Q[\mathbf{W}' \cup Anc(S)] = \sum_{\mathbf{W} \setminus \mathbf{W}'} Q[\mathbf{W} \cup Anc(S)].$$

Note that $\mathbf{W}' \cup Anc(S)$ in an ancestral set in $\mathcal{G}[\mathbf{W} \cup Anc(S)]$, and $\mathbf{W} \setminus \mathbf{W}' = (\mathbf{W} \cup Anc(S)) \setminus (\mathbf{W}' \cup Anc(S))$. Hence, Lemma B.1 concludes the proof.

$\square$

Suppose $\mathbf{H} \subseteq \mathbf{V}_{\mathrm{NS}}$ and let $\mathbf{H}_1, \ldots, \mathbf{H}_k$ denote the s-components of $\mathbf{H}$ in $\mathcal{G}^{\mathrm{s}}$. Then,

- $Q^{\mathrm{s}}[\mathbf{H}]$ decomposes as

$$Q^{\mathrm{s}}[\mathbf{H}] = Q^{\mathrm{s}}[\mathbf{H}_1]Q^{\mathrm{s}}[\mathbf{H}_2]\ldots Q^{\mathrm{s}}[\mathbf{H}_k].$$

- Let $m$ be the number of variables in $\mathbf{H}$, and consider a topological ordering of the variables in graph $\mathcal{G}^{\mathrm{s}}[\mathbf{H}]$, denoted as $V_{h_1} < \cdots < V_{h_m}$. Let $\mathbf{H}^{(0)} = \emptyset$ and for each $1 \le i \le m$, $\mathbf{H}^{(i)}$ denote the set of variables in $\mathbf{H}$ ordered before $V_{h_i}$ (including $V_{h_i}$). For every $1 \le j \le k$, $Q^{\mathrm{s}}[\mathbf{H}_j]$ can be computed from $Q^{\mathrm{s}}[\mathbf{H}]$ by

$$Q^{\mathrm{s}}[\mathbf{H}_j] = \prod_{\{i | V_{h_i} \in \mathbf{H}_j\}} \frac{Q^{\mathrm{s}}[\mathbf{H}^{(i)}]}{Q^{\mathrm{s}}[\mathbf{H}^{(i-1)}]}, \tag{15}$$

where $Q^{\mathrm{s}}[\mathbf{H}^{(i)}]$s can be computed by

$$Q^{\mathrm{s}}[\mathbf{H}^{(i)}] = \sum_{\mathbf{H} \setminus \mathbf{H}^{(i)}} Q^{\mathrm{s}}[\mathbf{H}]. \tag{16}$$

*Proof.* **First part:** according to the definition of $Q^{\mathrm{s}}[.]$, we have

$$Q^{\mathrm{s}}[\mathbf{H}] = P_{\mathbf{V}_{\mathrm{NS}} \setminus \mathbf{H}}(\mathbf{H}|Anc(S)) = \frac{P_{\mathbf{V}_{\mathrm{NS}} \setminus \mathbf{H}}(\mathbf{H}, Anc(S))}{P_{\mathbf{V}_{\mathrm{NS}} \setminus \mathbf{H}}(Anc(S))} = \frac{Q[\mathbf{H} \cup Anc(S)]}{Q[Anc(S)]}.$$

Note the last equality holds because $Anc(S)$ is an ancestral set in $\mathcal{G}$. Now, we have

$$Q[Anc(S)] = P_{\mathbf{V} \setminus Anc(S)}(Anc(S)) = P(Anc(S)) = P_{\mathbf{V}_{\mathrm{NS}} \setminus \mathbf{H}}(Anc(S)).$$

Similarly, for each $i \in [1:k]$, we have

$$Q^{\mathrm{s}}[\mathbf{H}_i] = \frac{Q[\mathbf{H}_i \cup Anc(S)]}{Q[Anc(S)]}$$

Therefore, the above equations imply that

$$Q^{\mathrm{s}}[\mathbf{H}] = Q^{\mathrm{s}}[\mathbf{H}_1]Q^{\mathrm{s}}[\mathbf{H}_2]\ldots Q^{\mathrm{s}}[\mathbf{H}_k] \iff \frac{Q[\mathbf{H} \cup Anc(S)]}{Q[Anc(S)]} = \frac{Q[\mathbf{H}_1 \cup Anc(S)]}{Q[Anc(S)]} \times \cdots \times \frac{Q[\mathbf{H}_k \cup Anc(S)]}{Q[Anc(S)]}$$
(17)

Let $\mathbf{C}_i$'s be the corresponding c-component of $\mathbf{H}_i$ (i.e., $\mathbf{H}_i \subseteq \mathbf{C}_i$ and $\mathbf{C}_i$ is a c-component of $\mathbf{H}_i \cup Anc(S)$). According to the definition of $\mathbf{C}_i$ and Corollary B.3 we have

$$Q[\mathbf{H}_i \cup Anc(S)] = Q[\mathbf{C}_i]Q[Anc(S) \setminus \mathbf{C}_i].$$

Moreover, since $\mathbf{C}_i$ is a c-component of $Anc(S) \cup \mathbf{H}_i$, then $\mathbf{C}_i \setminus \mathbf{H}_i$ should be union of some c-components of $Anc(S)$. Therefore, for each $i$, Corollary B.3 implies

$$Q[Anc(S)] = Q[\mathbf{C}_i \setminus \mathbf{H}_i]Q[Anc(S) \setminus (\mathbf{C}_i \setminus \mathbf{H}_i)] = Q[\mathbf{C}_i \setminus \mathbf{H}_i]Q[Anc(S) \setminus \mathbf{C}_i]$$

Putting the above equations together, we have

$$Q^{\text{s}}[\mathbf{H}_i] = \frac{Q[\mathbf{H}_i \cup Anc(S)]}{Q[Anc(S)]} = \frac{Q[\mathbf{C}_i]Q[Anc(S) \setminus \mathbf{C}_i]}{Q[\mathbf{C}_i \setminus \mathbf{H}_i]Q[Anc(S) \setminus \mathbf{C}_i]} = \frac{Q[\mathbf{C}_i]}{Q[\mathbf{C}_i \setminus \mathbf{H}_i]}. \tag{18}$$

Hence,

$$\prod_{i=1}^{k} \frac{Q[\mathbf{H}_i \cup Anc(S)]}{Q[Anc(S)]} = \prod_{i=1}^{k} \frac{Q[\mathbf{C}_i]}{Q[\mathbf{C}_i \setminus \mathbf{H}_i]}.$$

Consequently, we have

$$Q^{\text{s}}[\mathbf{H}] = Q^{\text{s}}[\mathbf{H}_1]Q^{\text{s}}[\mathbf{H}_2]\ldots Q^{\text{s}}[\mathbf{H}_k] \iff Q[\mathbf{H} \cup Anc(S)] = \frac{Q[Anc(S)]}{\prod_i Q[\mathbf{C}_i \setminus \mathbf{H}_i]}\prod_{i=1}^{k} Q[\mathbf{C}_i].$$

Note that $\mathbf{C}_i \setminus \mathbf{H}_i$ are disjoint subsets of $Anc(S)$, where each of them is the union of some c-components of $Anc(S)$. If $\mathbf{C}_{k+1} := Anc(S) \setminus \bigcup_i \mathbf{C}_i$, then Corollary B.3 implies the following.

$$Q[Anc(S)] = \prod_{i=1}^{k} Q[\mathbf{C}_i \setminus \mathbf{H}_i]Q[\mathbf{C}_{k+1}] \implies \frac{Q[Anc(S)]}{\prod_{i=1}^{k} Q[\mathbf{C}_i \setminus \mathbf{H}_i]} = Q[\mathbf{C}_{k+1}]$$

Finally, applying Corollary B.3 for $\mathbf{H} \cup Anc(S)$, we have

$$Q[\mathbf{H} \cup Anc(S)] = Q[\mathbf{C}_{k+1}]\prod_{i=1}^{k} Q[\mathbf{C}_i].$$

This concludes the first part.

**Second part:** the proof is similar to proof of Lemma B.2. Define $\mathbf{C} := \mathbf{H} \cup Anc(S)$. Let $\mathbf{C}_1, \ldots, \mathbf{C}_m$ be c-components of $C$ (w.l.o.g. assume that $\mathbf{H}_i \subseteq \mathbf{C}_i$ for $i \in [1:k]$). Consider a topological order of $V_{h_1} < V_{h_2} < \cdots < V_{h_n}$ for $\mathbf{H}$, and $V_{s_1} < \cdots < V_{s_{n'}}$ for $Anc(S)$. Since any $V_{h_i}$ is not an ancestor of $Anc(S)$, $V_{s_1} < \cdots < V_{s_{n'}} < V_{h_1} < V_{h_2} < \cdots < V_{h_n}$ is a topological order for $\mathbf{H} \cup Anc(S)$. According to Lemma B.2, we have

$$Q[\mathbf{C}_j] = \prod_{\{i|V_{c_i} \in \mathbf{C}_j\}} \frac{Q[\mathbf{C}^{(i)}]}{Q[\mathbf{C}^{(i-1)}]},$$

where $(c_1, c_2, \ldots, c_{n+n'}) = (s_1, \ldots, s_{n'}, h_1, \ldots, h_n)$. For each $i \in [1:n]$, let $\mathbf{H}^i := \{V_{h_1}, V_{h_2}, \ldots, V_{h_i}\}$ and $\mathbf{H}^0 = \varnothing$. For each $i > 0$, we have

$$\frac{Q[\mathbf{C}^{(i+n')}]}{Q[\mathbf{C}^{(i+n'-1)}]} = \frac{Q[\mathbf{H}^i \cup Anc(S)]}{Q[\mathbf{H}^{(i-1)} \cup Anc(S)]} = \frac{\frac{Q[\mathbf{H}^i \cup Anc(S)]}{Q[Anc(S)]}}{\frac{Q[\mathbf{H}^{(i-1)} \cup Anc(S)]}{Q[Anc(S)]}} = \frac{Q^{\text{s}}[\mathbf{H}^i]}{Q^{\text{s}}[\mathbf{H}^{(i-1)}]},$$

where the last equality holds according to Lemma B.4. It suffices to show that

$$Q^{\text{s}}[\mathbf{H}_j] = \prod_{\{i|V_{h_i} \in \mathbf{H}_j\}} \frac{Q^{\text{s}}[\mathbf{H}^{(i)}]}{Q^{\text{s}}[\mathbf{H}^{(i-1)}]}.$$

Equation 18 implies that

$$Q^{\text{s}}[\mathbf{H}_j] = \frac{Q[\mathbf{C}_j]}{Q[\mathbf{C}_j \setminus \mathbf{H}_j]}.$$

Hence, according to this equation and Lemma B.2, we have

$$Q[\mathbf{C}_j \setminus \mathbf{H}_j]Q^{\text{s}}[\mathbf{H}_j] = Q[\mathbf{C}_j] = \prod_{\{i|V_{c_i} \in \mathbf{C}_j\}} \frac{Q[\mathbf{C}^{(i)}]}{Q[\mathbf{C}^{(i-1)}]}$$

$$= \prod_{\{i|V_{c_i} \in \mathbf{C}_j \cap Anc(S)\}} \frac{Q[\mathbf{C}^{(i)}]}{Q[\mathbf{C}^{(i-1)}]} \prod_{\{i|V_{h_i} \in \mathbf{H}_j\}} \frac{Q^{\text{s}}[\mathbf{H}^{(i)}]}{Q^{\text{s}}[\mathbf{H}^{(i-1)}]}$$

$$= \prod_{\{i|V_{s_i} \in \mathbf{C}_j \setminus \mathbf{H}_j\}} \frac{Q[\mathbf{C}^{(i)}]}{Q[\mathbf{C}^{(i-1)}]} \prod_{\{i|V_{h_i} \in \mathbf{H}_j\}} \frac{Q^{\text{s}}[\mathbf{H}^{(i)}]}{Q^{\text{s}}[\mathbf{H}^{(i-1)}]}. \tag{19}$$

Based on the definition $\mathbf{C}_j$ and $\mathbf{H}_j$, $\mathbf{C}_j \setminus \mathbf{H}_j$ is the union of some C-components of $Anc(S)$. Denote the c-components of $\mathbf{C}_j \setminus \mathbf{H}_j$ by $A_1, \ldots, A_t$. Applying Lemma B.2 to $\mathbf{C} = Anc(S)$ with topological order $V_{s_1} < \cdots < V_{s_{n'}}$, we have

$$Q[A_1] = \prod_{\{i|V_{s_i} \in A_1\}} \frac{Q[\mathbf{C}^{(i)}]}{Q[\mathbf{C}^{(i-1)}]}$$

$$\vdots$$

$$Q[A_t] = \prod_{\{i|V_{s_i} \in A_t\}} \frac{Q[\mathbf{C}^{(i)}]}{Q[\mathbf{C}^{(i-1)}]}$$

By multiplying all $Q[A_i]$, we have

$$Q[\mathbf{C}_j \setminus \mathbf{H}_j] = Q[A_1 \cup A_2 \cdots \cup A_t] = \prod_{t'=1}^{t} Q[A_{t'}] = \prod_{\{i|V_{s_i} \in \mathbf{C}_j \setminus \mathbf{H}_j\}} \frac{Q[\mathbf{C}^{(i)}]}{Q[\mathbf{C}^{(i-1)}]}.$$

By substituting this in Equation 19, we obtain

$$Q^{\mathrm{s}}[\mathbf{H}_j] = \prod_{\{i|V_{h_i} \in \mathbf{H}_j\}} \frac{Q^{\mathrm{s}}[\mathbf{H}^{(i)}]}{Q^{\mathrm{s}}[\mathbf{H}^{(i-1)}]}.$$

To complete the proof, it suffices to show that $Q^{\mathrm{s}}[\mathbf{H}^{(i)}]$ are computable from $Q^{\mathrm{s}}[\mathbf{H}]$. Since we have used the topological order, $\mathbf{H}^i$ is an ancestral set in $\mathcal{G}[\mathbf{H}]$. Therefore, Lemma 4.4 implies that

$$Q^{\mathrm{s}}[\mathbf{H}^i] = \sum_{\mathbf{H} \setminus \mathbf{H}^i} Q^{\mathrm{s}}[\mathbf{H}].$$

This shows that $Q^{\mathrm{s}}[\mathbf{H}_i]$ is uniquely computable from $Q^{\mathrm{s}}[\mathbf{H}]$. $\qquad\square$

**Theorem 5.1.** *For disjoint subsets $\mathbf{X}$ and $\mathbf{Y}$ of $\mathbf{V}$, let*

$$\mathbf{X}_{\mathrm{AS}} := \mathbf{X} \cap \mathbf{V}_{\mathrm{AS}}, \quad \mathbf{X}_{\mathrm{NS}} := \mathbf{X} \cap \mathbf{V}_{\mathrm{NS}}, \text{ and } \mathbf{Y}_{\mathrm{NS}} := \mathbf{Y} \cap \mathbf{V}_{\mathrm{NS}}.$$

1. *Conditional causal effect $P_{\mathbf{X}}^{\mathrm{s}}(\mathbf{Y})$ is s-ID in $\mathcal{G}^{\mathrm{s}}$ if and only if*

$$(\mathbf{X}_{\mathrm{AS}} \perp\!\!\!\perp \mathbf{Y}|\mathbf{X}_{\mathrm{NS}}, S)_{\mathcal{G}^{\mathrm{s}}_{\underline{\mathbf{X}_{\mathrm{AS}}}\overline{\mathbf{X}_{\mathrm{NS}}}}}, \tag{20}$$

   *and $P_{\mathbf{X}_{\mathrm{NS}}}^{\mathrm{s}}(\mathbf{Y}, \mathbf{X}_{\mathrm{AS}})$ is s-ID in $\mathcal{G}^{\mathrm{s}}$.*

2. *Suppose $\mathbf{D} := Anc_{\mathcal{G}^{\mathrm{s}}[\mathbf{V}_{\mathrm{NS}} \setminus \mathbf{X}_{\mathrm{NS}}]}(\mathbf{Y}_{\mathrm{NS}})$ and let $\{\mathbf{D}_i\}_{i=1}^{k}$ denote the s-components of $\mathbf{D}$ in $\mathcal{G}^{\mathrm{s}}$. Conditional causal effect $P_{\mathbf{X}_{\mathrm{NS}}}^{\mathrm{s}}(\mathbf{Y}, \mathbf{X}_{\mathrm{AS}})$ is s-ID in $\mathcal{G}^{\mathrm{s}}$ if there are no s-Hedge in $\mathcal{G}^{\mathrm{s}}$ for any of $\{\mathbf{D}_i\}_{i=1}^{k}$.*

*Proof.* We first show that $(\mathbf{X}_{\mathrm{AS}} \perp\!\!\!\perp \mathbf{Y}|\mathbf{X}_{\mathrm{NS}}, S)_{\mathcal{G}^{\mathrm{s}}_{\underline{\mathbf{X}_{\mathrm{AS}}}\overline{\mathbf{X}_{\mathrm{NS}}}}}$ is a necessary condition. Suppose $(\mathbf{X}_{\mathrm{AS}} \not\perp\!\!\!\perp \mathbf{Y}|\mathbf{X}_{\mathrm{NS}}, S)_{\mathcal{G}^{\mathrm{s}}_{\underline{\mathbf{X}_{\mathrm{AS}}}\overline{\mathbf{X}_{\mathrm{NS}}}}}$. Denote by $\mathcal{G}'$ the equivalent DAG of $\mathcal{G}^{\mathrm{s}}$, obtained by adding the unobserved variables. Theorem 2 in [AMK24] shows that there are two SEMs $\mathcal{M}_1$ and $\mathcal{M}_2$ compatible with $\mathcal{G}'$ such that

$$P^{\mathcal{M}_1}(\mathbf{V}, \mathbf{U}|S = 1) = P^{\mathcal{M}_2}(\mathbf{V}, \mathbf{U}|S = 1), \text{and}$$
$$P_{\mathbf{X}}^{\mathcal{M}_1}(\mathbf{Y}|S = 1) \neq P_{\mathbf{X}}^{\mathcal{M}_2}(\mathbf{Y}|S = 1).$$

We use these SEMs to construct SCMs $\mathcal{M}_1'$ and $\mathcal{M}_2'$ compatible with $\mathcal{G}^{\mathrm{s}}$, in which we have

$$P^{\mathcal{M}_1'}(\mathbf{V}, \mathbf{U}|S = 1) = P^{\mathcal{M}_2'}(\mathbf{V}, \mathbf{U}|S = 1) \implies P^{\mathcal{M}_1'}(\mathbf{V}|S = 1) = P^{\mathcal{M}_2'}(\mathbf{V}|S = 1).$$

Note that all causal effects in both models are the same for $\mathcal{G}^{\mathrm{s}}$ and $\mathcal{G}$. Hence, when $(\mathbf{X}_{\mathrm{AS}} \not\perp\!\!\!\perp \mathbf{Y}|\mathbf{X}_{\mathrm{NS}}, S)_{\mathcal{G}^{\mathrm{s}}_{\underline{\mathbf{X}_{\mathrm{AS}}}\overline{\mathbf{X}_{\mathrm{NS}}}}}$ holds, then $P_{\mathbf{X}}^{\mathrm{s}}(\mathbf{Y})$ is not s-ID. It shows that this condition is a necessary condition.

Now suppose that $(\mathbf{X}_{\mathrm{AS}} \perp\!\!\!\perp \mathbf{Y}|\mathbf{X}_{\mathrm{NS}}, S)_{\mathcal{G}_{\underline{\mathbf{X}_{\mathrm{AS}}}\overline{\mathbf{X}_{\mathrm{NS}}}}}$ holds. According to Rule 2 of do-calculus, we have

$$P_{\mathbf{X}}(\mathbf{Y}|S = 1) = P_{\mathbf{X}_{\mathrm{NS}}}(\mathbf{Y}|\mathbf{X}_{\mathrm{AS}}, S = 1)$$

The above equation shows that $P_{\mathbf{X}}(\mathbf{Y}|S = 1)$ is s-ID in $\mathcal{G}^{\mathrm{s}}$ if and only if $P_{\mathbf{X}_{\mathrm{NS}}}(\mathbf{Y}|\mathbf{X}_{\mathrm{AS}}, S = 1)$ is s-ID in $\mathcal{G}^{\mathrm{s}}$. Moreover,

$$P_{\mathbf{X}_{\mathrm{NS}}}(\mathbf{Y}|\mathbf{X}_{\mathrm{AS}}, S = 1) = \frac{P_{\mathbf{X}_{\mathrm{NS}}}(\mathbf{Y}, \mathbf{X}_{\mathrm{AS}}|S = 1)}{P_{\mathbf{X}_{\mathrm{NS}}}(\mathbf{X}_{\mathrm{AS}}|S = 1)}.$$

Since $\mathbf{X}_{\mathrm{NS}} \cap \mathbf{V}_{\mathrm{AS}} = \varnothing$, acccording to Rule 3 of do-calculus, we have $P_{\mathbf{X}_{\mathrm{NS}}}(\mathbf{X}_{\mathrm{AS}}|S = 1) = P(\mathbf{X}_{\mathrm{AS}}|S = 1)$. Hence,

$$P_{\mathbf{X}_{\mathrm{NS}}}(\mathbf{Y}|\mathbf{X}_{\mathrm{AS}}, S = 1) = \frac{P_{\mathbf{X}_{\mathrm{NS}}}(\mathbf{Y}, \mathbf{X}_{\mathrm{AS}}|S = 1)}{P(\mathbf{X}_{\mathrm{AS}}|S = 1)}. \tag{21}$$

This shows that s-Identifiability of $P_{\mathbf{X}_{\mathrm{NS}}}(\mathbf{Y}|\mathbf{X}_{\mathrm{AS}}, S = 1)$ and $P_{\mathbf{X}_{\mathrm{NS}}}(\mathbf{Y}, \mathbf{X}_{\mathrm{AS}}|S = 1)$ are equivalent (note that $P(\mathbf{X}_{\mathrm{AS}}|S = 1) > 0$ due to the positivity assumption in Definition 4.1).

Let $\mathbf{W} := \mathbf{V}_{\mathrm{AS}} \setminus (\mathbf{X}_{\mathrm{AS}} \cup \mathbf{Y}_{\mathrm{AS}})$, then

$$P_{\mathbf{X}_{\mathrm{NS}}}(\mathbf{Y}|\mathbf{X}_{\mathrm{AS}}, S = 1) = \sum_{\mathbf{W}} P_{\mathbf{X}_{\mathrm{NS}}}(\mathbf{Y}, \mathbf{W}|\mathbf{X}_{\mathrm{AS}}, S = 1)$$

$$= \sum_{\mathbf{W}} P_{\mathbf{X}_{\mathrm{NS}}}(\mathbf{Y}_{\mathrm{AS}}, \mathbf{W}|\mathbf{X}_{\mathrm{AS}}, S = 1) P_{\mathbf{X}_{\mathrm{NS}}}(\mathbf{Y}_{\mathrm{NS}}|\mathbf{X}_{\mathrm{AS}}, \mathbf{Y}_{\mathrm{AS}}, \mathbf{W}, S = 1)$$

$$= \sum_{\mathbf{W}} P_{\mathbf{X}_{\mathrm{NS}}}(\mathbf{Y}_{\mathrm{AS}}, \mathbf{W}|\mathbf{X}_{\mathrm{AS}}, S = 1) P_{\mathbf{X}_{\mathrm{NS}}}(\mathbf{Y}_{\mathrm{NS}}|\mathbf{V}_{\mathrm{AS}}, S = 1).$$

The above equalities have been obtained by a marginalization over $\mathbf{W}$, chain rule, and replacing the definition of $\mathbf{W}$, respectively. According to Rule 3 of do-calculus, since $\mathbf{X}_{\mathrm{NS}} \cap \mathbf{V}_{\mathrm{AS}} = \varnothing$, we have

$$P_{\mathbf{X}_{\mathrm{NS}}}(\mathbf{Y}_{\mathrm{AS}}, \mathbf{W}|\mathbf{X}_{\mathrm{AS}}, S = 1) = P(\mathbf{Y}_{\mathrm{AS}}, \mathbf{W}|\mathbf{X}_{\mathrm{AS}}, S = 1).$$

Moreover, according to Lemma B.5, we have

$$P_{\mathbf{X}_{\mathrm{NS}}}(\mathbf{Y}_{\mathrm{NS}}|\mathbf{V}_{\mathrm{AS}}, S = 1) = \sum_{\mathbf{D} \setminus \mathbf{Y}_{\mathrm{NS}}} Q^{\mathrm{s}}[\mathbf{D}_1] \ldots Q^{\mathrm{s}}[\mathbf{D}_k].$$

Combining the aforementioned equations with Equation (21), we get

$$P_{\mathbf{X}_{\mathrm{NS}}}(\mathbf{Y}, \mathbf{X}_{\mathrm{AS}}|S = 1) = \sum_{\mathbf{W}} P(\mathbf{X}_{\mathrm{AS}}, \mathbf{Y}_{\mathrm{AS}}, \mathbf{W}|S = 1) \sum_{\mathbf{D} \setminus \mathbf{Y}_{\mathrm{NS}}} Q^{\mathrm{s}}[\mathbf{D}_1] \ldots Q^{\mathrm{s}}[\mathbf{D}_k]. \tag{22}$$

Now, note that according to Lemma B.6, if there is not any s-Hedge for $\mathbf{D}_i$s, then Function sID-Single will compute $Q^{\mathrm{s}}[\mathbf{D}_i]$s. Therefore, we can compute $P_{\mathbf{X}_{\mathrm{NS}}}(\mathbf{Y}, \mathbf{X}_{\mathrm{AS}}|S = 1)$ using the above equation, which concludes the proof. As a result, if Equation (5) holds, we have

$$P_{\mathbf{X}}^{\mathrm{s}}(\mathbf{Y}) = P_{\mathbf{X}_{\mathrm{NS}}}^{\mathrm{s}}(\mathbf{Y}|\mathbf{X}_{\mathrm{AS}}) = \sum_{\mathbf{W}} P(\mathbf{Y}_{\mathrm{AS}}, \mathbf{W}|\mathbf{X}_{\mathrm{AS}}, S = 1) \sum_{\mathbf{D} \setminus \mathbf{Y}_{\mathrm{NS}}} Q^{\mathrm{s}}[\mathbf{D}_1] \ldots Q^{\mathrm{s}}[\mathbf{D}_k]. \tag{23}$$

□

**Theorem 6.1.** *For disjoint subsets $\mathbf{X}$ and $\mathbf{Y}$ of $\mathbf{V}$, $P_{\mathbf{X}}(\mathbf{Y})$ can be uniquely computed from $P^{\mathrm{s}}(\mathbf{V})$ in the augmented ADMG $\mathcal{G}^{\mathrm{s}}$ if and only if*

$$(\mathbf{Y} \perp\!\!\!\perp S|\mathbf{X})_{\mathcal{G}_{\overline{\mathbf{X}}}^{\mathrm{s}}},$$

*and $P_{\mathbf{X}}^{\mathrm{s}}(\mathbf{Y})$ is s-ID in $\mathcal{G}^{\mathrm{s}}$.*

*Proof.* If $(\mathbf{Y} \not\perp\!\!\!\perp S|\mathbf{X})_{\mathcal{G}_{\overline{\mathbf{X}}}}$, then [BT15, Theorem 2] implies that $P_{\mathbf{X}}(\mathbf{Y})$ is not s-recoverable. If $(\mathbf{Y} \perp\!\!\!\perp S|\mathbf{X})_{\mathcal{G}_{\overline{\mathbf{X}}}}$, then according to Rule 1 of do-calculus, we have

$$P_{\mathbf{X}}(\mathbf{Y}|S = 1) = P_{\mathbf{X}}(\mathbf{Y}). \tag{24}$$

Therefore, the identifiability of $P_{\mathbf{X}}(\mathbf{Y})$ is equivalent to $P_{\mathbf{X}}(\mathbf{Y}|S = 1)$, which concludes the proof.
□

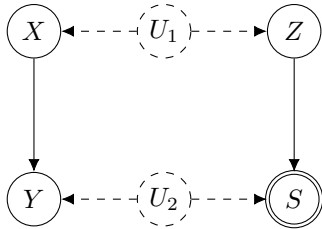

Figure 7: A DAG, where $U_1$ and $U_2$ are unobservable, and $S$ represents the auxiliary variable for modeling a sub-population.

## D  Numerical Experiment

We conduct a numerical experiment to demonstrate the significance of the s-ID problem and assess the output of Algorithm 1. Note that the experiment is simple and can be run on a system with any level of computational power.

Consider the following structural causal model (SCM) with the causal graph depicted in Figure 7.

$$
\begin{aligned}
U_1 &\sim Bern(0.5) \\
U_2 &\sim Bern(0.5) \\
X &= U_1 \oplus \varepsilon_x, \ \varepsilon_x \sim Bern(0.2) \\
Y &= X \oplus U_2 \\
Z &= U_1 \oplus \varepsilon_z, \ \varepsilon_z \sim Bern(0.2)
\end{aligned}
$$

Herein, $\oplus$ denotes the XOR operation, all the variables $\{U_1, U_2, \varepsilon_x, \varepsilon_z, \varepsilon_{s_1}, \varepsilon_{s_2}, \varepsilon_{s_3}\}$ are independent, and $Bern(p)$ denotes a Bernoulli random variable with parameter $p$. Now, consider the sub-population with the following mechanism:

$$
S = (Z \times \varepsilon_{s_1}) \oplus (U_2 \times \varepsilon_{s_2}) \oplus \varepsilon_{s_3}, (\varepsilon_{s_1}, \varepsilon_{s_2}, \varepsilon_{s_3}) \sim (Bern(0.6), Bern(0.9), Bern(0.1))
$$

We now consider the problem of estimating the causal effect of $X$ on $Y$ in this sub-population. Particularly, our goal is to calculate $P_{X=0}(Y = 1|S = 1)$.

### D.1  Theoretical Analysis

To analyze and compare the s-ID and ID algorithms, we first determine the exact values of $P_{X=0}(Y = 1|S = 1)$ and $P_{X=0}(Y = 1)$. According to the equation of $Y$ in the SCM, we have

$$
P_{X=x}(Y = y) = P_{X=x}(y = x \oplus U_2) = P(U_2 = y \oplus x).
$$

Since $U_2$ is a Bernoulli random variable with parameter 0.5, the above probability always equals 0.5. For $P_X(Y|S = 1)$, we have

$$
P_{X=x}(Y = y|S = 1) = P_{X=x}(Y = x \oplus U_2|S = 1) = P(U_2 = x \oplus y|S = 1).
$$

Therefore, for $X = 0$ and $Y = 1$, we need to compute $P(U_2 = 1|S = 1)$. By using the equation of $S$ in the model,

$$
\begin{aligned}
P(U_2 = 1|S = 1) &= \frac{P(U_2 = 1, S = 1)}{P(U_2 = 1, S = 1) + P(U_2 = 0, S = 1)} \\
&= \frac{P(U_2 = 1)P(S = 1|U_2 = 1)}{P(U_2 = 0)P(S = 1|U_2 = 0) + P(U_2 = 1)P(S = 1|U_2 = 1)}
\end{aligned}
$$

Note that $P(U_2 = 0) = P(U_2 = 1) = 0.5$, hence, we have

$$
P(U_2 = 1|S = 1) = \frac{P(S = 1|U_2 = 1)}{P(S = 1|U_2 = 0) + P(S = 1|U_2 = 1)}
$$

Since $\varepsilon_{s_1}$ and $\varepsilon_{s_3}$ and $Z$ are independent variables, let $W$ be $Z \times \varepsilon_{s_1} \oplus \varepsilon_{s_3}$; then, we have

$$W \sim Bern(0.1 \times (1 - 0.6 \times 0.5) + 0.9 \times 0.5 \times 0.6) = Bern(0.34).$$

Note that $W$ and $U_2 \times \varepsilon_{s_2}$ are independent, and $S = W + U_2 \times \varepsilon_{s_2}$; thus,

$$P(S = 1|U_2 = 0) = P(W = 1) = 0.34$$
$$P(S = 1|U_2 = 1) = P(W = 1)P(\varepsilon_{s_2} = 0) + P(W = 0)P(\varepsilon_{s_2} = 1)$$
$$= 0.34 \times 0.1 + 0.66 \times 0.9 = 0.628$$

Hence,

$$P_{X=0}(Y = 1|S = 1) = \frac{0.628}{0.628 + 0.34} \approx 0.648 \tag{25}$$

### D.2 Empirical Analysis

The ID algorithm returns the following simple expression for $P_X(Y)$

$$P_X(Y) = P(Y|X). \tag{26}$$

On the other hand, the s-ID algorithm returns

$$P_X(Y|S = 1) = \sum_Z P(Y|X, Z, S = 1)P(Z|S = 1). \tag{27}$$

Next, we generated 3000 samples from the population. We then computed $S$ for each generated sample and collected the samples where $S = 1$, resulting in 1469 available samples from our target sub-population. Recall that our goal was to estimate $P_{X=0}(Y = 1|S = 1)$. Consider the following two approaches.

- If we consider the s-ID algorithm and the existence of the selection bias $S$, applying the simple plug-in estimator to the formula in (27) results in

$$P_{X=0}(Y = 1|S = 1) \approx \hat{P}(Y = 1|X = 0, Z = 0)\hat{P}(Z = 0)$$
$$+ \hat{P}(Y = 1|X = 0, Z = 1)\hat{P}(Z = 1) = 0.641 \tag{28}$$

- Suppose we ignore the presence of sub-population and apply the ID algorithm. In this scenario, we have to estimate $P_X(Y)$ using the empirical distribution of variables, i.e., $\hat{P}(\mathbf{V})$. If we use the ID algorithm, we need to estimate the quantity mentioned in equation (26). The empirical estimate for this case is

$$P_X(Y) \approx \hat{P}(Y|X) = 0.725. \tag{29}$$

A comparison between the estimation results of our proposed method in Equation 28 and the classical ID problem in Equation (29) with the true underlying value in Equation (25) shows that our approach accurately computes the target causal effect. On the other hand, ignoring the subtleties related to sub-population can lead to erroneous estimation.

