# OpenReview forum: "Causal Effect Identification in a Sub-Population with Latent Variables"
_NeurIPS.cc/2024/Conference — NeurIPS 2024 poster_

### Official Review · Reviewer_YSpJ · 2024-06-27

**Soundness:** 3
**Presentation:** 2
**Contribution:** 3
**Rating:** 6
**Confidence:** 3

**Summary:**

This paper addresses the S-ID (Sub-population Identification) problem in causal inference, extending it to scenarios with latent variables. The S-ID problem seeks to determine if a causal effect in a specific sub-population can be uniquely computed from observational data pertaining to that sub-population. The authors introduce new graphical definitions such as S-components and S-Hedges, which are extensions of classical notions like C-components and Hedges. They present a sufficient graphical condition for determining if a causal effect is S-ID and propose a sound algorithm for solving the S-ID problem in the presence of latent variables. Additionally, they show a reduction from the S-Recoverability problem to the S-ID problem.

**Strengths:**

Technical quality: The paper presents thorough theoretical analysis, including formal definitions, lemmas, examples and theorems, as well as the reduction derivation.

**Weaknesses:**

Empirical evaluation: there is no experiments at all besides the last section in appendix briefly describling how the authors want to conduct them. That is to say, this paper lacks experimental results or real-world case studies to demonstrate the practical applicability and performance of the proposed solution, especiallly for the two recursive algorithms
Comparison to other existing methods: The paper surely follows the id, c-id, S-Recoverability literature for related work, but still could benefit from a more extensive comparison with other latent variable models in causal inference, [1-3] to name a few

[1] Liu, Yuhang, et al. "Identifying weight-variant latent causal models." arXiv preprint arXiv:2208.14153 (2022).

[2] Sherman, Eli, and Ilya Shpitser. "Identification and estimation of causal effects from dependent data." Advances in neural information processing systems 31 (2018).

[3] Kocaoglu, Murat, Karthikeyan Shanmugam, and Elias Bareinboim. "Experimental design for learning causal graphs with latent variables." Advances in Neural Information Processing Systems 30 (2017).

**Questions:**

1. In Remark 5.3.  the authors "conjects that this algorithm is also complete". Is there any example or situation that it returns a false negative?
2. The reduction from S-Recoverability to S-ID in Section 6 seems to suggest that S-ID is a more general problem. Is there any scenario where solving S-ID would be more useful than solving the S-Recoverability problem?

**Limitations:**

The paper proved that both proposed algorithms are sound for s-id, but did not show any guarantee for the completeness. Nevertheless, this limitation is mentioned in conclusion.

---

> ### Author Rebuttal · Authors · 2024-08-06
>
> We thank the reviewer for their comments. We first answer the two questions and then discuss the reviewer's concerns mentioned in the weakness section.
>
> ------
>
> > **Q1**: Is there any example or situation that it returns a false negative?
>
> We have not found any examples where the algorithm produces a false negative. In fact, we have evidence to support that the s-ID algorithm always returns "Fail" when the target causal effect is not s-ID and have observed no false negatives. That is why we conjectured that the algorithm is complete. We have tried to prove this conjecture, and we have succeeded in proving it for some specific cases. However, proving it in general has eluded us so far (perhaps due to the complexity of s-component and s-Hedge structures and the positivity constraint).
>
> ----
>
> > **Q2**: Is there any scenario where solving S-ID would be more useful than solving the S-Recoverability problem?
>
> There are two main implications of Theorem 6.1 and Remark 6.2:
>
> - **s-ID is a more general problem**:  As the reviewer correctly points out, s-ID is more general than s-Recoverability in the sense that an algorithm for s-ID can tackle s-Recoverability problem as well.
>
> - **s-ID is a more practical setting**: Recall that given the observational distribution of a sub-population, s-Recoverability attempts to identify $P_{X}(Y)$ (the causal effect over the entire population), while s-ID attempts to identify $P_{X}(Y | S = 1)$ (the causal effect over the sub-population). Not surprisingly s-Recoverability does not succeed except under very special settings.
> This is what Remark 6.2 argues: the condition for s-recoverability to successfully identify $P_{X}(Y | S = 1)$ is quite restrictive, whereas s-ID has a more realistic condition.
> Accordingly, there are indeed many examples where $P_{X}(Y | S = 1)$ is identifiable, but $P_{X}(Y)$ is not. For instance, please refer to Figure 4, where our algorithm returns
>
> $ P_{X_1}(Y | S = 1) = \sum_{Z_1, Z_2} P^s (Z_1, Z_2) \sum_{X_2} P^s (X_2 | X_1, Z_1, Z_2) \sum_{\tilde{X}_1} P^s(Y | \tilde{X}_1, X_2, Z_1, Z_2) P^s (\tilde{X}_1 | Z_1, Z_2).$
>
> While s-Recoverability fails to recover $P_{X_1}(Y)$ since this effect is indeed non-identifiable.
>
> ------
>
> ## Regarding empirical evaluation
>
> Computing a causal effect numerically typically involves several steps:
>
> - **Causal Discovery (a.k.a., Causal Identification)**: Learning the causal graph from the available data.
>
> - **Causal Effect Identification**: This step focuses on determining whether the causal effect is identifiable and, if so, deriving a formula for it.
>
> - **Numerical Estimation**: Given a finite set of samples, compute the expression derived in the identification phase numerically.
>
> - **Evaluation**: This step measures and evaluates the quality of estimators proposed in the estimation step, which often includes sensitivity analysis.
>
>
> Similar to [1-5], this paper focuses on addressing the identification part. Still, we provided an example with real-world variables in Example 1 and also carried out a simple numerical experiment in Appendix C using a finite set of samples. However, designing an end-to-end pipeline is beyond the scope of this paper and is an interesting future research direction. This would involve collecting a real-world dataset, conducting rigorous causal discovery, designing a proper estimator based on the variable distribution, and applying various evaluations.
>
>
> [1] Tian, Jin, and Judea Pearl. "A general identification condition for causal effects." In Proceedings of the Eighteenth National Conference on Artificial Intelligence (2002)
>
> [2] Shpitser, Ilya, and Judea Pearl. "Identification of conditional interventional distributions." Proceedings of the 22nd Conference on Uncertainty in Artificial Intelligence, 2006.
>
> [3] Bareinboim, Elias, Jin Tian, and Judea Pearl. "Recovering from selection bias in causal and statistical inference." In Proceedings of the Twenty-Eighth AAAI Conference on Artificial Intelligence, pp. 2410-2416. 2014.
>
> [4] Lee, Sanghack, Juan D. Correa, and Elias Bareinboim. "General identifiability with arbitrary surrogate experiments." In Uncertainty in artificial intelligence, pp. 389-398. PMLR, 2020.
>
> [5] Jaber, Amin, Adele Ribeiro, Jiji Zhang, and Elias Bareinboim. "Causal identification under Markov equivalence: calculus, algorithm, and completeness." Advances in Neural Information Processing Systems 35 (2022).
>
> ------
>
> > Comparison to other existing methods: The paper surely follows the id, c-id, S-Recoverability literature for related work, but still could benefit from a more extensive comparison with other latent variable models in causal inference, [1-3] to name a few...
>
> Thank you for mentioning these works, especially Paper [2], which is more relevant to our problem.
> Papers [1] and [3], however, seem to be more focused on causal discovery, which is the first step of the pipeline we described earlier. Nonetheless, as the reviewer suggested, we will include more related work.
>
> -----
>
> Finally, we noticed that the reviewer is a bit concerned about the "Soundness" and "Presentation" of the paper. We would appreciate it if the reviewer could provide further details so that we can improve the paper.

---

> > ### Comment · Reviewer_YSpJ · 2024-08-08
> >
> > Having read the authors' rebuttal and comments from other reviewers, my questions have been adequately addressed. As a result, I increase my evalutaion to 6.

---

### Official Review · Reviewer_4gRi · 2024-07-11

**Soundness:** 3
**Presentation:** 3
**Contribution:** 3
**Rating:** 7
**Confidence:** 3

**Summary:**

This paper extends the sub-population causal effect identifiability (S-ID) problem to include latent variables by adapting classical graphical definitions such as connected-components and Hedges. It proposes a sound algorithm to compute causal effects in sub-populations with latent variables.

**Strengths:**

1. The paper is written well and easy to understand.
2. Examples in each section helps to understand the underlying idea easily.
3. All the necessary background is discussed clearly.

**Weaknesses:**

1. Example 1 could be a better one because socioeconomic status can cause cardiovascular disease.
2. It would be good to include a sub section for summarising any assumptions made.
3. It would be good to include some real-world use-cases benefiting from such setting of causal effect identification.

**Questions:**

See weaknesses section

**Limitations:**

Limitations are discussed

---

> ### Author Rebuttal · Authors · 2024-08-06
>
> We appreciate the reviewer's suggestions and positive feedback.
>
> ---
>
> > Example 1 could be a better one because socioeconomic status can cause cardiovascular disease.
>
> We acknowledge that the causal graph might not be completely accurate - the primary goal of this example is to demonstrate the difference between ID and s-ID algorithms over a simple graph.
> For the particular edge that the reviewer is mentioning (from socioeconomic status to cardiovascular disease), if we simplify/replace socioeconomic status with income, it might be fair to assume that this effect is mediating through $X$, which is the medication choice.
>
> ----
>
> > It would be good to include a sub section for summarising any assumptions made.
>
> As the reviewer suggested, we will add a section that summarizes the problem setup and the corresponding assumptions. We will also add a table of notations to improve the clarity.
>
> ----
>
> > It would be good to include some real-world use-cases benefiting from such setting of causal effect identification.
>
> Thank you for your suggestion. While our main contribution is to establish the theoretical framework for addressing the s-ID problem in the presence of latent variables, we agree that providing more real-world examples in addition to Example 1 would be valuable for readers looking to apply our proposed method. Therefore, we will include another example in the final version.

---

> > ### Comment · Reviewer_4gRi · 2024-08-12
> > **Thank you for your response**
> >
> > I thank the authors for their response. I've read their response and I will stay with my score.

---

### Official Review · Reviewer_APnQ · 2024-07-12

**Soundness:** 3
**Presentation:** 3
**Contribution:** 3
**Rating:** 7
**Confidence:** 2

**Summary:**

This paper extends the S-ID problem, which asks if a causal effect within a specific sub-population can be identified using only observational data from that group. The authors consider the scenarios where some variables are latent. They provide a  sufficient graphical condition to determine whether a causal effect is S-ID and propose an algorithm based on this criterion. While the paper proves the algorithm's soundness, it suggests it might also be complete. Finally, they show that solving S-ID can solve a related problem called S-Recoverability.

**Strengths:**

- The paper addresses an important problem in causal inference.
- The paper is very well-written and covers the prerequisites very well.

**Weaknesses:**

See the Questions section below.

**Questions:**

- Although the work provides rigorous theoretical contributions, it would have been nice to evaluate how it would also work empirically, especially in a close-to-real-world scenario.
- There are many variables and notations used throughout the paper. A table summarizing these notations and their definitions could improve readability.
- How often do real-world problems satisfy the restrictive condition in Equation (8)​​?

**Limitations:**

The limitations of the proposed approach have not been discussed.

---

> ### Author Rebuttal · Authors · 2024-08-06
>
> We thank the reviewer and appreciate the positive feedback on the importance of the problem and the clarity of the presentation.
> Below we answer the questions.
>
> ---
>
> > Although the work provides rigorous theoretical contributions, it would have been nice to evaluate how it would also work empirically, especially in a close-to-real-world scenario.
>
> Computing a causal effect numerically typically involves several steps:
>
> - **Causal Discovery (a.k.a., Causal Identification)**: Learning the causal graph from the available data.
>
> - **Causal Effect Identification**: This step focuses on determining whether the causal effect is identifiable and, if so, deriving a formula for it.
>
> - **Numerical Estimation**: Given a finite set of samples, compute the expression derived in the identification phase numerically.
>
> - **Evaluation**: This step measures and evaluates the quality of estimators proposed in the estimation step, which often includes sensitivity analysis.
>
>
> Similar to [1-5], this paper focuses on addressing the identification part. Still, we provided an example with real-world variables in Example 1 and also carried out a simple numerical experiment in Appendix C using a finite set of samples. However, designing an end-to-end pipeline is beyond the scope of this paper and is an interesting future research direction. This would involve collecting a real-world dataset, conducting rigorous causal discovery, designing a proper estimator based on the variable distribution, and applying various evaluations.
>
>
> [1] Tian, Jin, and Judea Pearl. "A general identification condition for causal effects." In Proceedings of the Eighteenth National Conference on Artificial Intelligence (2002)
>
> [2] Shpitser, Ilya, and Judea Pearl. "Identification of conditional interventional distributions." Proceedings of the 22nd Conference on Uncertainty in Artificial Intelligence, 2006.
>
> [3] Bareinboim, Elias, Jin Tian, and Judea Pearl. "Recovering from selection bias in causal and statistical inference." In Proceedings of the Twenty-Eighth AAAI Conference on Artificial Intelligence, pp. 2410-2416. 2014.
>
> [4] Lee, Sanghack, Juan D. Correa, and Elias Bareinboim. "General identifiability with arbitrary surrogate experiments." In Uncertainty in artificial intelligence, pp. 389-398. PMLR, 2020.
>
> [5] Jaber, Amin, Adele Ribeiro, Jiji Zhang, and Elias Bareinboim. "Causal identification under Markov equivalence: calculus, algorithm, and completeness." Advances in Neural Information Processing Systems 35 (2022).
>
>
> ------
>
> > There are many variables and notations used throughout the paper. A table summarizing these notations and their definitions could improve readability.
>
> Thank you for your suggestion. We will add a table summarizing our key notations.
>
> > How often do real-world problems satisfy the restrictive condition in Equation (8)​​?
>
> The condition in Equation (8) specifies that following an intervention on $X$, the target variable(s) $Y$ should have the same distribution in both the sub-population and the entire population. In other words, after the intervention, $Y$ should not be impacted by the sub-population, which is a very stringent requirement.
> For instance, in the family of graphs where $Y$ is a parent of $S$, Equation (8) does not hold. Consequently, $P_{X}(Y)$ is not identifiable from $P (\mathbf{V} | S = 1)$, while $P_{X}(Y | S = 1)$ can still be identifiable from $P (\mathbf{V} | S = 1)$.

---

### Official Review · Reviewer_rYoX · 2024-07-17

**Soundness:** 3
**Presentation:** 4
**Contribution:** 4
**Rating:** 7
**Confidence:** 3

**Summary:**

The paper presents a sound algorithm for checking the s-identifiability of causal effects under sub-populations. This work complements earlier work on s-ID by generalizing the causal graphs to allow hidden confounders (no causal sufficiency). Specifically, the paper introduces the notions of s-components and s-Hedge, in parallel to the classical notions of c-components and c-hedge, and derives theoretical results based on those. The main theorem (Theorem 5.1) summarizes the condition under which a causal effect is s-ID, and a detailed algorithm is also proposed for deriving an identifying formula. Moreover, a reduction from s-recoverability to s-ID is mentioned which provides yet another approach to solve the s-recoverability problem.

**Strengths:**

- I think the problem is quite meaningful since selection bias can be common in the data collection process.
- It is great that the paper not only introduces novel notions such as s-components and s-hedges but also thoroughly reviews the classical notions of c-components and c-hedges, so we can make a comparison. The lemmas and theorems are also in parallel (but different ) to the previous ones for classical identification in [Tian, Pearl], which makes these profound concepts easier to penetrate.
- I found the examples helpful, especially Examples 4 and 6, in aiding my understanding of definitions.
- In general, a hard work that contains valuable theoretical contributions.

**Weaknesses:**

- It seems that the s-ID method is sound but not complete, but I guess the paper already contains enough contributions and the completeness part can always be the future work.
- More intuitions can be provided on the difference between s-ID and ID at the end of page 2 - it would be helpful to provide a more intuitive explanation for Example 1 (in addition to an explanation based on identifying formulas) for readers to see the importance of the problem.
- The definition of $Q[]$ seems to be ambiguous. In Section 2 last subsection, $Q[X]$ is defined as the interventional distribution $Q[X] := P_{x} (V \setminus X)$, but in Theorem 3.4 $Q[D]$ seems to mean $Pr_x(D)$. It may be helpful to clarify the formal definition of $Q[D]$.

**Questions:**

- Are there any insights on the difference between having $P^s(V)$ vs. having $P(V)$ for identifiabilty? For example, would more variables become dependent so they now belong to the same s-components when collecting data under $P^s(V)$?
- Is there any evidence (counterexample) proving that Algorithm 1 is not complete?
- Regarding the reduction from s-recoverability to s-ID, I'm wondering if there is any impact of this reduction besides theoretical interests. For example, will the reduction-based approach for s-recoverability be more computationally efficient?

**Limitations:**

OK.

---

> ### Author Rebuttal · Authors · 2024-08-06
>
> We thank the reviewer for the detailed feedback and appreciate the positive comments on the contributions and presentation of the paper.
>
> -----
>
> > Are there any insights on the difference between having $P^s(V)$ vs. having $P(V)$ for identifiabilty? For example, would more variables become dependent so they now belong to the same s-components when collecting data under $P^s(V)$?
>
> We confirm your intuition. The presence of selection bias ($S$) injects additional dependencies among variables. Consequently, the rules of do-calculus cannot be directly applied to $P^s$ because of the influence of $S$. As a result, the ID algorithm is no longer applicable to the s-ID setting.
> To tackle this challenge, we defined s-components over the non-ancestors of $S$ to capture these additional dependencies among c-components.
> We further defined s-Hedge based on the s-component and derived our identification results based on that.
>
> -----
>
> > Is there any evidence (counterexample) proving that Algorithm 1 is not complete?
>
> We have not found any examples where the algorithm produces a false negative. In fact, we have evidence to support that the s-ID algorithm always returns "Fail" when the target causal effect is not s-ID. This suggests that there are no false negatives and that the algorithm is complete. We have tried to prove this conjecture, and we have succeeded in proving it for some specific cases. However, proving it in the general case has evaded us so far due to the complexity of s-component and s-Hedge structures and the positivity constraint. This is definitely a future work direction for us, but also for the community at large.
>
> ----
>
> > Regarding the reduction from s-recoverability to s-ID, I'm wondering if there is any impact of this reduction besides theoretical interests. For example, will the reduction-based approach for s-recoverability be more computationally efficient?
>
>
> The goal of our reduction was to demonstrate that the s-ID algorithm can also address the s-Recoverability problem, thus, algorithm 2 might not be the optimal choice for the s-recoverability problem. We also note that computational complexity is not a limitation for either approach. However, in cases where the causal effect is identifiable, our method could provide a different expression. Having different expressions for the target causal effect could potentially be beneficial for developing an estimator that numerically computes the causal effects using a finite set of samples. This can be further studied in future work.
>
> ----
> > More intuitions can be provided on the difference between s-ID and ID at the end of page 2 - it would be helpful to provide a more intuitive explanation for Example 1 (in addition to an explanation based on identifying formulas) for readers to see the importance of the problem.
>
> We thank the reviewer for the suggestion. We will include a more detailed discussion in the revised version, where we have an additional page.
>
> ----
>
> ### Regarding the definition of $Q[]$:
>
> Thank you for pointing out this typo - we will fix it.
> In graph with observable variables $\mathbf{V}$, for each $\mathbf{D} \subseteq \mathbf{V}$ we have $Q[\mathbf{D}] = P_{\mathbf{V} \setminus \mathbf{D}}(\mathbf{D})$. $Q[\mathbf{D}]$ follows this definition in Theorem 3.4.

---

### Decision · Program_Chairs · 2024-09-25

**Decision:**

Accept (poster)

**Comment:**

The authors presents a sound algorithm for checking s-identifiability of causal effects under sub-populations given the causal structure and partially observed data. The unanimous recommendation from reviewers is accept, which I concur with and recommend.